# Structure of SHOC2-KRAS-PP1C complex reveals RAS isoform-specific determinants and insights into targeting complex assembly by RAS inhibitors

RAF activation is essential for MAPK signaling and is mediated by RAS binding and the dephosphorylation of a conserved phosphoserine by the SHOC2–RAS–PP1C complex. MRAS forms a high-affinity SHOC2–MRAS–PP1C (SMP) complex, while canonical RAS isoforms (KRAS, HRAS, NRAS) form analogous but lower-affinity assemblies. Yet, cancers driven by oncogenic KRAS, HRAS, or NRAS remain strongly SHOC2-dependent, suggesting that these weaker complexes contribute to tumorigenesis. To elucidate how canonical RAS proteins form lower-affinity ternary complexes, the cryo-EM structure of the SHOC2–KRAS–PP1C (SKP) complex stabilized by Noonan syndrome mutations is described. The SKP architecture is similar to the SMP complex but forms fewer contacts and buries less surface area due to the absence of MRAS-specific structural features in KRAS that enhance complex stability. RAS inhibitors MRTX1133 and RMC-6236 alter Switch-I/II conformations, thereby blocking SKP assembly more effectively than they disrupt pre-formed complexes. These RAS inhibitors do not affect SMP formation because they do not bind MRAS. Since MRAS is upregulated in resistance to KRAS inhibition, we characterize a MRAS mutant capable of binding MRTX1133. This MRAS mutant can form an SMP complex, but MRTX1133 blocks its assembly, demonstrating the feasibility of dual SKP and SMP targeting. Overall, our findings define isoform-specific differences in SHOC2–RAS–PP1C complex formation and support a strategy to prevent both SKP and SMP assemblies to overcome resistance in RAS-driven cancers.

The RAS–RAF–MEK–ERK signaling cascade transmits mitogenic signals from the cell surface to the nucleus, orchestrating key cellular programs such as proliferation, differentiation, and survival[1]. Dysregulation of this pathway, often through mutations in RAS or RAF, contributes to approximately 20–30% of human cancers[2]. Activation begins when receptor tyrosine kinases stimulate GDP-to-GTP exchange on RAS proteins, enabling them to recruit RAF kinases to the plasma membrane. RAF then undergoes conformational changes and

dimerization[1,3], activating downstream MEK and ERK kinases through phosphorylation[1,4].

RAF activation is tightly regulated by phosphorylation at conserved serine residues in conserved regions 2 (CR2-pS) and 3 (CR3-pS), which flank the kinase domain[5]. In the inactive state, these sites are bridged by a 14-3-3 dimer, stabilizing RAF in an autoinhibited conformation[6–8]. Upon GTP loading, membrane-bound RAS engages the RAF RAS-binding domain (RBD) and cysteine-rich domain (CRD),

✉ e-mail: Dhirendra.Simanshu@nih.gov

facilitating RAF membrane localization and exposing CR2-pS for site-specific dephosphorylation by the SHOC2–RAS–PP1C ternary complex[9–11]. This dephosphorylation relieves autoinhibition and permits dimerization of RAF protomers, allowing a single 14-3-3 dimer to bridge CR3-pS sites across the dimer interface.

The SHOC2–RAS–PP1C ternary complex is composed of the leucine-rich repeat scaffold protein SHOC2, the catalytic phosphatase PP1C, and an active RAS isoform[12]. Our biophysical and structural studies, along with those of others, have shown that MRAS, which shares ~50% sequence identity with canonical RAS isoforms (KRAS, HRAS, and NRAS), forms a high-affinity SHOC2–MRAS–PP1C (SMP) complex and revealed the architecture of the ternary complex[13–16]. Canonical RAS isoforms form analogous but lower-affinity complexes. SHOC2 acts as a scaffolding protein to assemble the ternary complex and contains 20 leucine-rich repeats (LRRs) along with an N-terminal intrinsically disordered region harboring an RVxF PP1C-binding motif. PP1C, a serine/threonine phosphatase with three isoforms (PP1CA, PP1CB, PP1CC), specifically dephosphorylates RAF CR2-pS when activated by SHOC2 and GTP-bound RAS[12,13,16].

Despite their reduced binding affinity, canonical RAS containing SHOC2–H/K/NRAS–PP1C complexes remain functionally important in cancer. Notably, (i) the lower-affinity SHOC2–H/K/NRAS–PP1C complexes are still capable of dephosphorylating RAF CR2-pS substrates[17]; (ii) MRAS knockout mice exhibit no apparent phenotype, suggesting functional compensation by canonical RAS isoforms[18,19]; and (iii) oncogenic mutants of H/K/NRAS, particularly Q61 mutants, show strong dependence on SHOC2 for cancer cell growth and survival, indicating that SHOC2–PP1C complexes with oncogenic RAS can overcome weaker binding and effectively promote RAF dephosphorylation and activation[15,16,20,21]. However, the structural basis for ternary complex formation by canonical RAS isoforms remains unclear due to their biochemical instability. Recent advances in targeting KRAS include allele-specific inhibitors, such as sotorasib and adagrasib, which target the KRAS-G12C mutation[22,23]. However, resistance frequently arises through multiple mechanisms, including MRAS mutations and upregulation, emphasizing the need for strategies that simultaneously target KRAS and MRAS[24–27].

In this work, we use Noonan syndrome- (SHOC2-M173I and PP1CA-P50R) and cancer-associated mutations (KRAS-Q61R) to assemble a stabilized SHOC2–KRAS–PP1C (stabilized SKP) complex. The stabilized SKP complex shares a similar overall architecture with SMP but reveals isoform-specific contacts that account for the reduced binding affinity of KRAS relative to MRAS. We evaluate MRTX1133, a KRAS-G12D-selective inhibitor effective against both nucleotide states, and RMC-6236, a pan-RAS inhibitor that binds active RAS via cyclophilin A[28–30], as inhibitors of the formation of wild-type SHOC2–KRAS–PP1C (SKP) and SMP complexes. Both inhibitors bind KRAS with high affinity and prevent SKP assembly more potently than they disrupt preformed SKP complexes. Neither inhibitor binds MRAS or disrupts the SMP complex. An MRTX1133-sensitive MRAS variant that retains high-affinity binding to SHOC2 and PP1C is used to evaluate dual inhibition of SMP and SKP complexes, establishing proof-of-concept for dual targeting of MRAS and KRAS. These findings reveal RAS isoform-specific features of SHOC2 and PP1C complexes and suggest that disrupting both SKP and SMP may help limit MAPK pathway reactivation in RAS-driven cancers.

## Results

### SHOC2 dependency in RAS-mutant cancers and stabilized SKP complex

We and others have previously reported (summarized in Supplementary Table 1) distinct differences in binding affinity between SHOC2–PP1C complexes formed with MRAS versus those formed with the canonical RAS isoforms HRAS, KRAS, and NRAS by different biophysical techniques[13–16]. Among these, MRAS forms the highest-affinity

complex with SHOC2 and PP1C ($K_D \approx 120$ nM by SPR), while H/K/NRAS form analogous ternary complexes with significantly lower affinity ($K_D \approx 0.7$–4 µM by SPR)[13]. Despite this weaker binding, genome-wide CRISPR/Cas9 fitness screens from the Cancer Dependency Map (DepMap; https://depmap.org/portal/) revealed that cancer cell lines harboring oncogenic mutations (G12X, G13X, Q61X) in H/K/NRAS display a marked dependency on SHOC2 for proliferation and survival[15,16,21,31]. Paradoxically, no comparable dependency was observed for MRAS despite its higher affinity binding to SHOC2 and PP1C. Chronos score correlations further support these findings, showing strong co-dependency between SHOC2 and mutant RAS isoforms, particularly those with Q61 and G13 mutations, with a weaker correlation for G12 mutations. (Fig. 1a, b). These observations suggest that while the SHOC2–PP1C interaction with canonical RAS isoforms is biochemically less stable, it is functionally more critical in the context of RAS-driven cancers. This disconnect between affinity and cellular dependency may reflect a requirement for dynamic or transient SHOC2–H/K/NRAS–PP1C engagement to support efficient RAF dephosphorylation and sustained MAPK pathway activity in oncogenic settings.

To define the structural basis of canonical RAS interaction with SHOC2 and PP1C, we aimed to determine the structure of the SHOC2–KRAS–PP1C complex. Previous efforts using X-ray crystallography and cryo-electron microscopy (cryo-EM) were unsuccessful due to the dissociation of the low-affinity complex during sample preparation[16,32]. To overcome this challenge, we applied a rational mutagenesis approach to stabilize complex formation, building on prior insights into SHOC2–RAS–PP1C interactions (Fig. 1c). The SHOC2-M173I and PP1CA-P50R mutations, both associated with Noonan syndrome, had previously been shown to enhance SMP complex affinity by ~2–3 fold[13,33]. Additionally, the Noonan syndrome-associated MRAS-Q71R mutation, which introduces an arginine at the interface, strengthens SMP complex formation by forming additional polar contacts[15]. SMP and SKP affinities measured by ITC, conducted in 500 mM NaCl and 5% glycerol to maintain PP1C solubility, are higher (900 nM for SMP, 7 µM for SKP) than those measured earlier by SPR at 150 mM NaCl (120 nM for SMP, 0.7 µM for SKP), reflecting the effect of ionic strength rather than intrinsic affinity differences (see Methods for details). The corresponding cancer-associated mutation in KRAS (Q61R), when combined with Noonan syndrome mutations SHOC2-M173I and PP1CA-P50R, led to a substantial ~40-fold increase in SKP complex affinity, reducing the $K_D$ from 7 µM to 154 nM (Fig. 1d), which is tighter than the SMP complex affinity as measured by ITC under high salt conditions.

This stabilized SKP complex, in combination with optimized graphene oxide grids, enabled structure determination by cryo-EM at 3.0 Å resolution (Fig. 1e; Supplementary Fig. 1a–g and Supplementary Table 2). The resulting Coulomb potential map enabled detailed visualization of the molecular interfaces that stabilize the ternary complex and allowed direct comparison with the SMP complex, thereby providing insight into the conserved and isoform-specific features of SHOC2–RAS–PP1C assembly.

### Cryo-EM structure of the stabilized SKP complex

The overall architecture of the stabilized SKP complex reveals that SHOC2 adopts a curved, horseshoe-shaped structure, serving as a scaffold that brings KRAS and PP1CA into proximity, resembling previously determined SMP complex structures[13–16]. All three proteins—SHOC2, KRAS, and PP1CA—are well resolved in the cryo-EM Coulomb potential map (Fig. 1e). SHOC2 shows progressively lower local resolution toward the C-terminal end of its LRR domain, likely due to increased flexibility (Fig. 2a). The active site channels of manganese-bound PP1CA remain fully exposed within the stabilized SKP complex (Fig. 2b). SHOC2 engages PP1CA through an RVxF motif located in its N-terminal intrinsically disordered region, which binds to a conserved hydrophobic pocket on PP1CA used by other RVxF-containing regulatory proteins (Fig. 2b)[34].

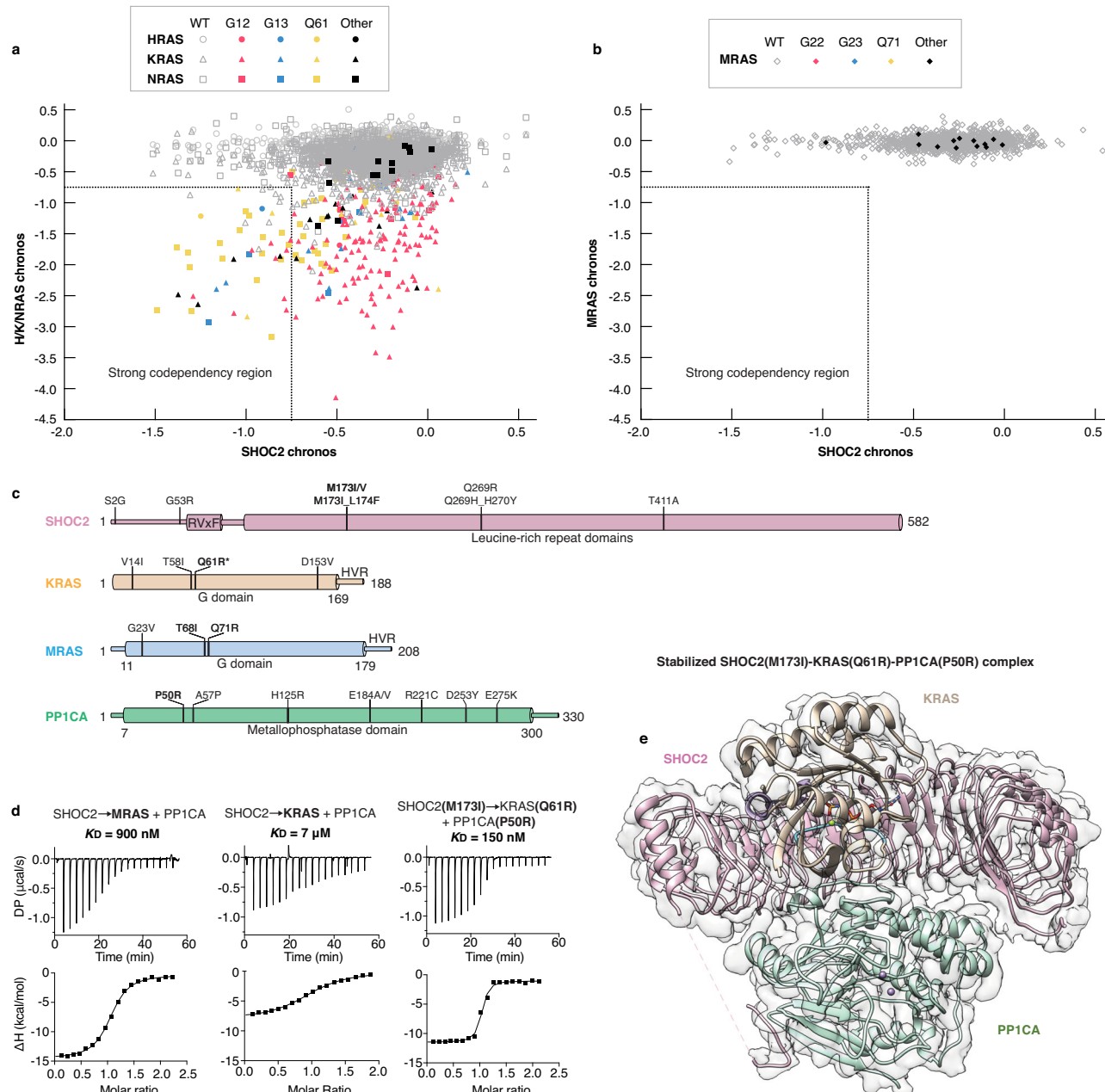

**Fig. 1 | Comparative dependency and assembly of the SKP complex. a** DepMap chronos scores for 1,178 cancer cell lines from the Q4 2024 release for SHOC2 versus the chronos score of HRAS, KRAS, and NRAS. Cell lines containing WT H/K/NRAS, oncogenic H/K/NRAS G12, G13, Q61, and other mutations are shown in gray, red, blue, yellow, and black shapes, respectively. **b** DepMap chronos scores for 1178 cancer cell lines from the Q4 2024 release for SHOC2 versus the chronos score of MRAS. Cell lines containing WT MRAS, oncogenic MRAS G22, G23, Q71, and other mutations are shown in gray, red, blue, yellow, and black diamonds, respectively. **c** Domain architecture of SHOC2, PP1CA, and the G-domains of MRAS and KRAS. Noonan syndrome mutations for each protein are noted. Noonan syndrome mutations only occur in the PP1CB isoform of PP1C and have been noted using

PP1CA numbering. Noonan syndrome mutation MRAS-Q71R equivalent in KRAS (KRAS-Q61R, denoted with *) is oncogenic in nature. Mutations used to stabilize the SKP complex are shown in bold. **d** ITC experiments measuring complex formation of SMP, SKP, and stabilized SKP using SHOC2-M173I, KRAS-Q61R, and PP1CA-P50R. Arrow indicates the titrant, SHOC2, being injected into the cell containing RAS and PP1CA. This nomenclature is used for all ITC traces. The $K_D$ is calculated from two technical replicates. **e** The 3.0 Å resolution DeepEMhancer sharpened map of the stabilized SKP complex is represented as a grey transparent surface. SHOC2, KRAS, and PP1CA are presented as cartoons with the same color described in Fig. 1c. Disordered regions are depicted as dashed lines.

The LRR domain of SHOC2 forms a curved solenoid, with its concave surface partially wrapped around PP1CA. The Coulomb potential map for 19 of the 20 LRRs is well resolved. KRAS, like other RAS isoforms, adopts a G-domain fold comprising a six-stranded β-sheet and five α-helices. Within the complex, the Switch-I (residues 30-38) and Switch-II (residues 60-76) regions of KRAS interact with the upper concave face of the LRR domain of SHOC2, while additional

contacts are made between KRAS and PP1CA through residues in the pre-Switch-I and interswitch regions (Fig. 2b). These interactions contribute to complex stability, burying a total of ~5200 Å² of surface area and forming 10 salt bridges and 17 hydrogen bonds.

Although all three pathogenic mutations enhanced complex affinity biochemically, only one forms a direct contact in the stabilized SKP structure. PP1CA-P50R engages in van der Waals interactions with

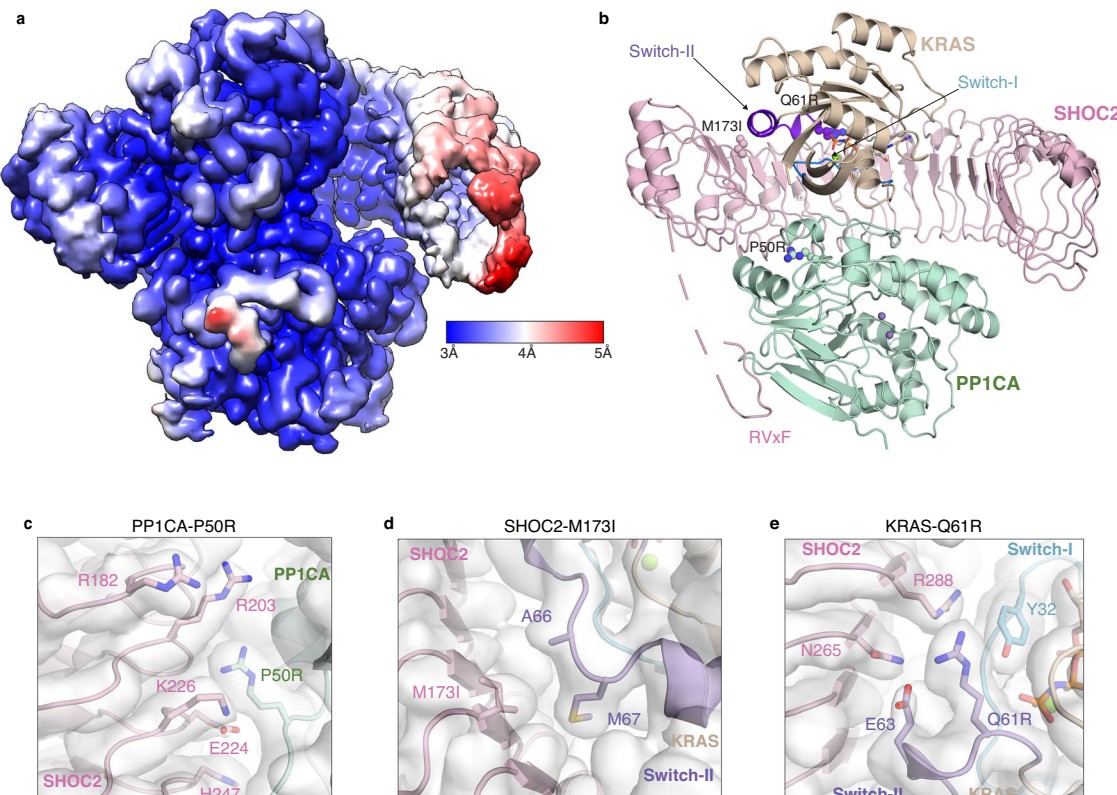

**Fig. 2 | Cryo-EM structure of the stabilized SKP complex and stabilizing interactions formed by pathogenic mutations. a** Local resolution DeepEMhancer sharpened map of the stabilized SKP complex colored from highest to lowest resolution (blue to red). **b** The overall structure of the stabilized SKP complex in cartoon representation (using the same colors described in Fig. 1c). Disordered regions are depicted as dashed lines. Enlarged views showing the interaction formed by the three Noonan syndrome mutations introduced to stabilize the SKP complex with DeepEMhancer sharpened map; **c** PP1CA-P50R, **d** SHOC2-M173I, and **e** KRAS-Q61R.

SHOC2 residues N202 and E224 (Fig. 2c). The SHOC2-M173I mutation, previously shown to increase SMP complex affinity, does not form direct contacts in the stabilized SKP complex (Fig. 2d). The substitution of methionine by isoleucine likely increases local hydrophobicity within the LRR domain, indirectly contributing to complex stabilization. Similarly, the KRAS-Q61R mutation does not make direct contacts at the KRAS–SHOC2 interface (Fig. 2e), unlike the MRAS-Q71R mutant SMP complex structure, which engages through both direct and water-mediated interactions with SHOC2 residues N265 and R288[15]. These water-mediated interactions may also be present in the stabilized SKP complex but are unresolved at the current cryo-EM resolution.

## Mechanistic insights into KRAS versus MRAS engagement with SHOC2 and PP1C

The stabilized SKP, and SMP complexes superimpose with a low root mean square deviation (RMSD) of 1.2 Å over Cα atoms, indicating a conserved overall architecture (Fig. 3a). Despite this similarity, the stabilized SKP complex buries approximately 1000 Å² less surface area and forms 6 fewer salt bridges and 12 fewer hydrogen bonds compared to the SMP complex as calculated by PISA[35] and PDBSum[36], consistent with the lower binding affinity of wild-type KRAS relative to MRAS for SHOC2 and PP1C (Figs. 1d and 3b, c and Supplementary Figs. 2a, b and 3). The most substantial loss occurs at the SHOC2–KRAS interface, which buries 450 Å² less surface area due to reduced contacts with the Switch-I and -II regions of KRAS (Fig. 3d). Only three hydrogen bonds are observed between SHOC2 and KRAS, involving KRAS residues E37 from Switch-I and E63 and Q70 from Switch-II, which interact with SHOC2 residues R177, N265, and D106, respectively. In contrast, the SHOC2–MRAS interface contains seven hydrogen bonds (Fig. 3d and Supplementary Fig. 4a).

A distinguishing feature of MRAS is the β5−α4 helical loop within its allosteric lobe, which forms van der Waals interactions and a hydrogen bond between MRAS-H132 and SHOC2-E428 (Fig. 3e). This loop is one residue shorter and more aliphatic in KRAS, resulting in the absence of equivalent contacts with SHOC2 (Fig. 3e and Supplementary Fig. 4b). The interswitch region also diverges significantly between KRAS and MRAS (Fig. 3f, g and Supplementary Fig. 4c). In KRAS, Q43 forms a single main-chain hydrogen bond with PP1CA-D179, while R41 makes minimal contact due to an internal hydrogen bond with E3 that reorients its side chain away from the PP1CA surface (Fig. 3f and Supplementary Fig. 4c). In contrast, MRAS-H53 (equivalent to KRAS-Q43) forms two hydrogen bonds through both backbone and side chain with PP1CA-D179, and MRAS-L51 (equivalent to KRAS-R41) engages in additional van der Waals interactions with PP1CA (Fig. 3g).

Smaller but notable losses (~300 Å² each) are also observed at the KRAS−PP1C and SHOC2−PP1C interfaces in the stabilized SKP complex. KRAS lacks the N-terminal extension present in MRAS, which in the SMP complex contributes van der Waals contacts and a hydrogen bond between MRAS-S4 and PP1CA-E218 (Fig. 3h). This extension is further stabilized by internal van der Waals interactions between MRAS residues P7 and W60 (Fig. 3h). Additionally, the SHOC2 RVxF motif and twelve flanking residues form a β-hairpin that engages PP1CA in the SMP complex. In the stabilized SKP structure, only the RVxF motif and five adjacent residues are resolved, and this region adopts a shorter β-strand conformation (Fig. 3a). The comparative analysis suggests that the cumulative loss of buried surface area, salt bridges, hydrogen bonds, and van der Waals contacts in the stabilized SKP complex is not due to the stabilization mutations but sequence differences between MRAS and KRAS. This underlies the reduced stability and lower affinity of the wild-type SKP complex relative to the SMP complex.

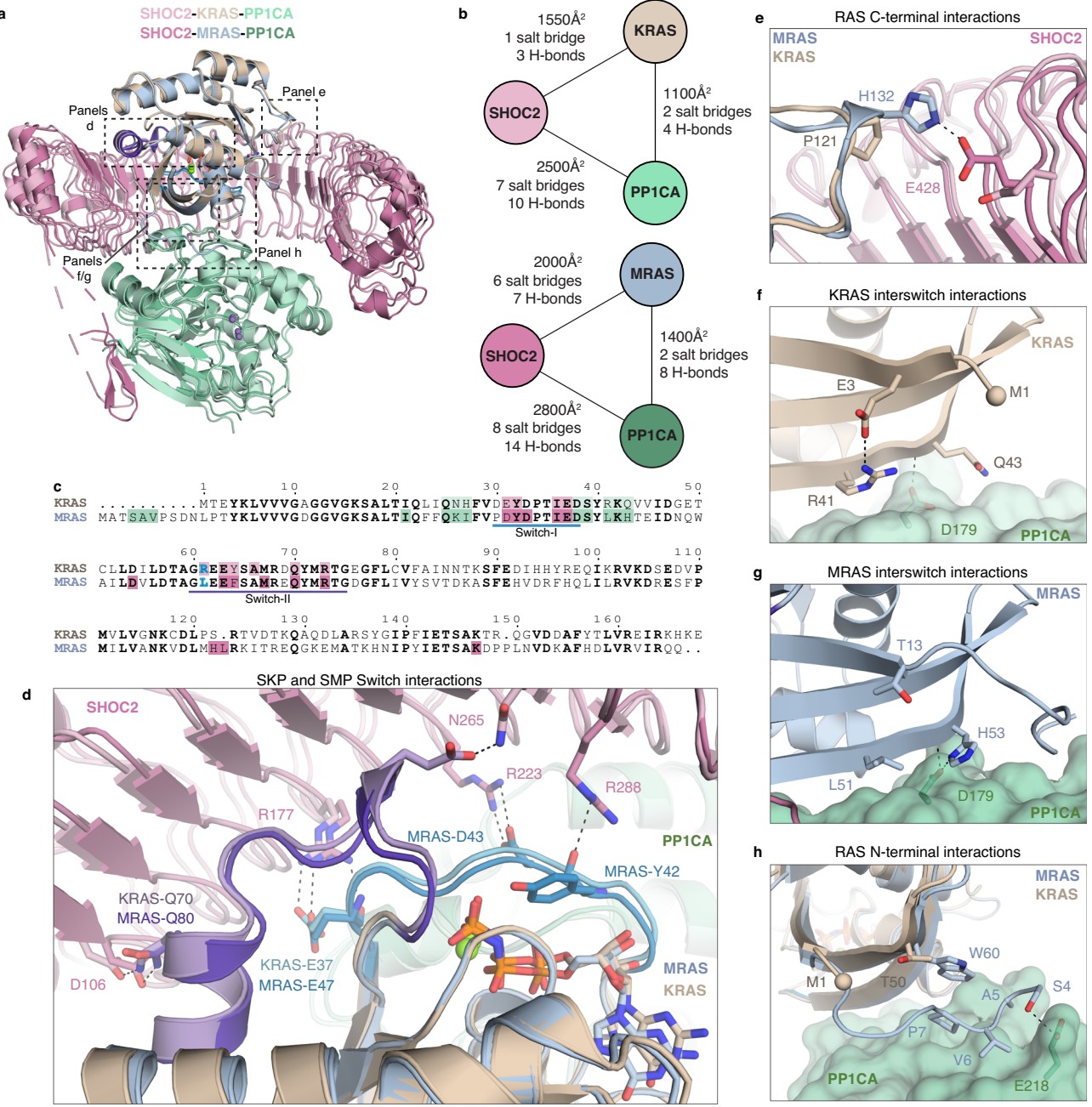

**Fig. 3 | KRAS- and MRAS-specific interactions in SHOC2−RAS−PP1C complexes.**
**a** Superposition of the stabilized SHOC2-KRAS-PP1CA complex (light pink, brown, and light green, respectively) with the SHOC2-MRAS-PP1CA complex (dark pink, blue, and dark green, respectively, PDB 7TVF). **b** Schematic comparing the interactions and contacts formed in the stabilized SKP and SMP complexes using PISA and PDBSum. **c** Sequence alignment of KRAS and MRAS showing residues interacting with SHOC2 (pink) and PP1CA (green) in the stabilized SKP and SMP complexes, respectively. Switch-I and -II are highlighted in blue and purple, respectively. The KRAS-Q61R and MRAS-Q71L mutations are shown in blue. **d** Interaction of Switch-I (light blue) and Switch-II (light purple) of KRAS (brown) with SHOC2 (light pink) and interaction of Switch-I (dark blue) and Switch-II (dark purple) of MRAS (blue) with SHOC2 (dark pink). **e** Interaction of MRAS β5-α4 helical loop (blue) with SHOC2 (dark pink). This loop in KRAS (brown) is shorter and fails to interact with SHOC2 (light pink). Interswitch engagement of **f** KRAS (brown) and **g** MRAS (blue) with PP1CA. **h** The N-terminal extension of MRAS (blue) interacts with PP1CA. KRAS (brown) lacks the N-terminal extension.

## Distinct switch-II conformations define effector-specific RAS interactions

Effector binding to RAS is essential for propagating downstream signaling and is typically mediated through its GTP-bound conformation, which exposes key interaction surfaces in the Switch-I and, in some cases, Switch-II regions. Switch-I adopts a conserved conformation across diverse effectors, including CRAF, PI3Kα, Rgl2, Sin1, SHOC2−PP1C, the allosteric site of RasGEF SOS1, and RasGAP NF1, optimized for high-affinity engagement (Fig. 4)[11,13,37–40].

In contrast, Switch-II is markedly more flexible in both GDP- and GTP-bound states and is often unresolved in unbound structures unless stabilized by crystal packing. Effector or regulator binding typically stabilizes this region, but its conformation varies depending on the binding partner. For example, in the KRAS−CRAF complex, Switch-II does not directly engage the RBD−CRD but is resolved as a flexible loop (residues 60−64) followed by a short α2 helix (residues 65−75) (Fig. 4a). In contrast, effectors such as Sin1, Rgl2, and RasGAP NF1 make direct contacts with Switch-II, leading to a shortened loop

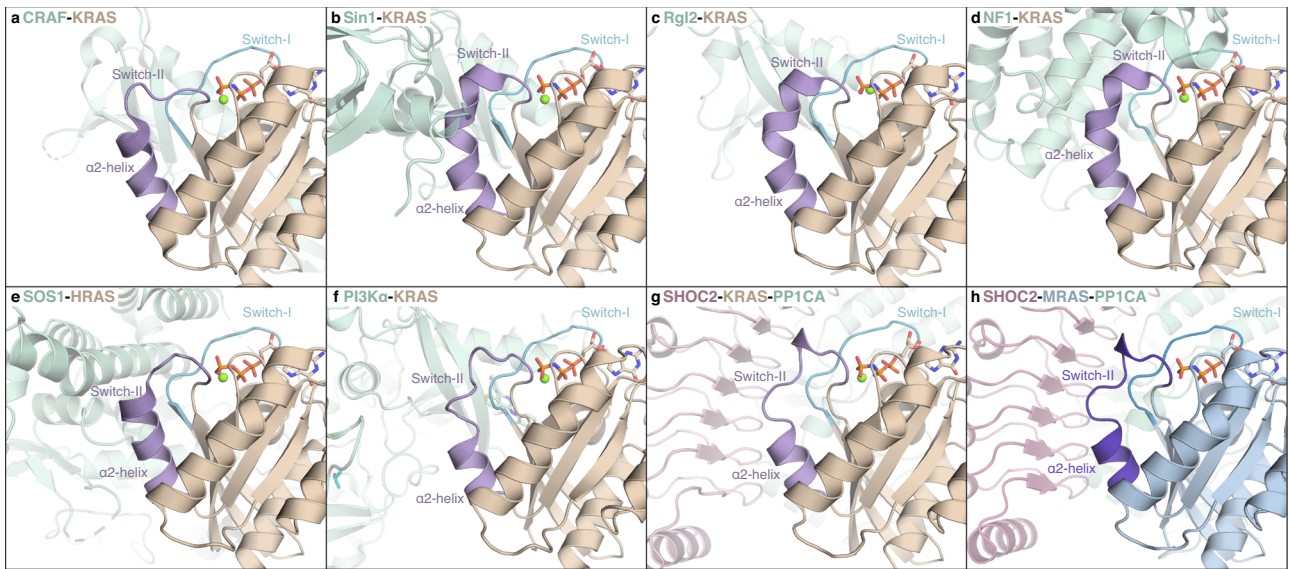

**Fig. 4 | Structural plasticity of Switch-II in RAS complexes with effector and regulatory proteins.** Switch-I (light blue) adopts a conserved conformation across effectors and regulators, while Switch-II (purple) displays diverse conformations depending on the binding partner. KRAS bound to CRAF (PDB 6XI7) (**a**), Sin1 (PDB 7LC1) (**b**), Rgl2 (PDB 8B69) (**c**), NF1(PDB 6OB2) (**d**), SOS1 (HRAS bound structure, PDB 1NVU) (**e**) show variable Switch-II conformation ranging from flexible loops to extended helices. KRAS in complex with PI3Kα (PDB 9C15) (**f**), and KRAS (**g**) and MRAS (**h**) in complex with SHOC2–PP1CA (PDBs: 7TVF, and this study) show extensive Switch-II engagement with a shortened α2 helix and extended loop. These comparisons highlight the adaptable nature of Switch-II, which assumes distinct conformations tailored to subsets of effector or regulator proteins.

(residue 60) and an elongated, bent α2 helix (residues 61–75) (Fig. 4b–d). A similar conformation is observed in HRAS bound to the allosteric site of RasGEF SOS1, with a slight variation in loop positioning compared to KRAS–CRAF (Fig. 4e)[41].

PI3Kα and SHOC2–PP1C, in both SKP and SMP complexes, engage Switch-II extensively, inducing a distinct conformation with a shortened α2 helix (residues 68–75 in KRAS; 78–85 in MRAS) and an extended loop (residues 60–67 in KRAS; 70–77 in MRAS) (Fig. 4f–h). These comparisons highlight the structural plasticity of Switch-II, which, rather than adopting a single conserved conformation, serves as a dynamic interface that assumes distinct structural states depending on the mode of interaction with subsets of effectors or regulators.

## Structural basis for MRTX1133-mediated inhibition of SKP complex assembly

KRAS-G12C inhibitors, such as sotorasib and adagrasib, function by covalently modifying the cysteine at position 12 in the GDP-bound state, thereby preventing nucleotide exchange and subsequent activation. However, resistance to these inhibitors frequently arises through the emergence of other active KRAS mutations[27,42]. More recently, adaptive resistance has been attributed to the mislocalization of Scribble (SCRIB) from the membrane-associated SHOC2–SCRIB–PP1C complex to the cytoplasm[24,25]. Cytoplasmic SCRIB inactivates the Hippo pathway, enabling YAP nuclear translocation and transcription of target genes, including MRAS. MRAS upregulation, along with nucleotide exchange via SOS, contributes to the feedback reactivation of MAPK signaling[24,26] and MRAS-regulated resistance pathways[43,44]. It is currently unknown whether MRTX1133 induces similar resistance mechanisms, though studies suggest it can trigger EGFR-mediated feedback and promote upregulation of RAS pathway genes such as *Kras, Yap1, Myc*, and *Cdk6/Abcb1a/b*[45–48]. These observations raise the possibility that co-targeting oncogenic KRAS and MRAS could prevent SHOC2–RAS–PP1C (SRP) complex–driven escape mechanisms.

To investigate whether pharmacological targeting of RAS proteins could disrupt the assembly of SHOC2–RAS–PP1C complexes, we evaluated the effects of RAS inhibitors on SKP and SMP formation. Given the importance of these complexes in MAPK signaling, including potential roles in adaptive resistance to KRAS inhibitors, we sought a proof-of-concept approach to identify small molecules capable of blocking both KRAS- and MRAS-mediated complex assembly.

MRTX1133 is a selective, non-covalent KRAS-G12D inhibitor that binds with high affinity to both GDP- and GTP-bound forms, engaging a pocket formed by Switch-II and the α3 helix[29]. MRTX1133 can also bind wild-type and other oncogenic mutants of KRAS, but with relatively lower affinity[49]. Using ITC (with 500 mM NaCl, see Methods for details), we confirmed that MRTX1133 binds both wild-type KRAS and KRAS-G12D with low nM affinity in both nucleotide states, albeit weaker than measured by SPR under lower salt conditions[29,49], with tighter binding to the GDP- than GMPPNP-bound state in both cases (Fig. 5a and Supplementary Fig. 5a).

To better understand the structural basis of MRTX1133 binding to wild-type KRAS and how this might impact complex formation, we determined the crystal structures of wild-type KRAS(GDP) and KRAS(GMPPNP) bound to MRTX1133 at 1.4 Å and 1.9 Å resolution, respectively (Fig. 5b and Supplementary Fig. 5b–d). These structures allow direct comparison with existing KRAS-G12D–MRTX1133 structures and provide a model system for analyzing inhibitor effects in the context of non-mutant SKP complex assembly. They superimpose with a low RMSD (0.3 Å) and show minor differences in Switch-I and -II conformations (Supplementary Fig. 5e). MRTX1133 occupies the same Switch-II pocket as in KRAS-G12D. In wild-type KRAS, the absence of an aspartate at position 12 is compensated by bridging water molecules and crystallization ions (e.g., sulfate or chloride) that help coordinate the P-loop and nucleotide with MRTX1133's bicyclic piperazine group (Supplementary Fig. 5f).

Despite its weaker binding to wild-type KRAS, MRTX1133 effectively blocked SKP complex assembly in ITC experiments and weakly dissociated pre-formed complexes (Fig. 5c and Supplementary Fig. 5g). Structural modeling revealed that MRTX1133 binding displaces the α2 helix and alters Switch-I, resulting in steric clashes with SHOC2-Y129 and impaired SHOC2 interaction (Fig. 5d). Thus, MRTX1133 binding to KRAS not only inhibits RAF binding but also prevents SKP complex formation.

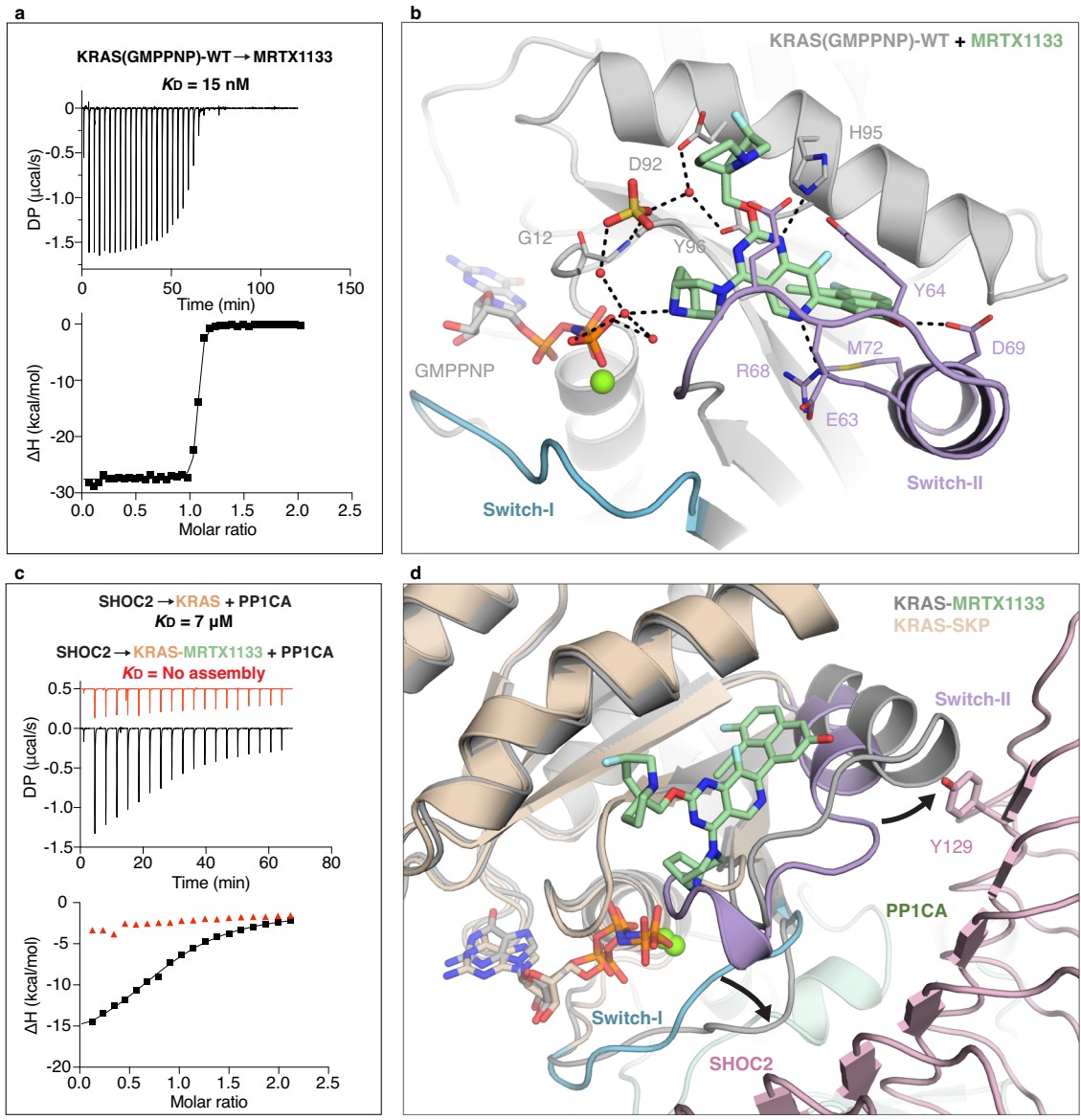

**Fig. 5 | MRTX1133 binding to KRAS impairs SKP formation through Switch-II rearrangement. a** ITC profile showing binding of MRTX1133 to GMPPNP-bound wild-type KRAS. The $K_D$ is calculated from two technical replicates. **b** Crystal structure of the wild-type KRAS(GMPPNP)-MRTX1133 with KRAS in gray and MRTX1133 in green. **c** ITC profiles showing SKP formation in the absence (black) and presence of MRTX1133 (red). The $K_D$ is calculated from two technical replicates.

**d** Superposition of the wild-type KRAS(GMPPNP)-MRTX1133 complex (gray) with KRAS from the SKP complex (brown). Black arrows indicate conformational changes in KRAS switch regions upon MRTX1133 binding. Switch-II in the inhibitor-bound conformation clashes with SHOC2 residue Y129 (pink sticks), explaining inhibition of complex formation.

## Proof-of-concept for dual inhibition of KRAS and MRAS to disrupt ternary complex assembly

To explore broader inhibition of RAS isoforms, we tested the pan-RAS inhibitor RMC-6236, which forms a tight binary complex with cyclophilin A (CypA) and then binds H/K/NRAS proteins, blocking Switch-I/II engagement with effectors[28]. ITC measurements showed that RMC-6236 binds CypA with a $K_D$ of 108 nM and forms a ternary complex with KRAS at 127 nM (Fig. 6a and Supplementary Fig. 6a). Like MRTX1133, RMC-6236–CypA disrupted and prevented SKP complex formation, through blocking of Switch-I engagement with SHOC2 and CypA occupying the SHOC2-binding site (Fig. 6b, c and Supplementary Fig. 6b). However, it did not bind MRAS or interfere with SMP formation, suggesting that resistance to RMC-6236 could similarly arise through MRAS upregulation (Fig. 6a, b and Supplementary Fig. 6b).

KRAS and MRAS share 53% sequence identity, though only six residues within the MRTX1133 Switch-II binding pocket differ between

the two proteins: D21A, F74Y (the only residue that also forms van der Waal interactions in the stabilized SKP, and SMP structures), E79D, R105H, F106Y, and L109Q (MRAS numbering) (Fig. 7a and Supplementary Fig. 7a). As such, MRTX1133 does not bind MRAS in either GDP- or GMPPNP-bound states (Fig. 7b and Supplementary Fig. 7b), nor does it affect SMP complex assembly (Supplementary Fig. 7c). While ligand redesign could be explored to accommodate MRAS, we pursued a simpler approach: introducing mutations into MRAS Switch-II pocket to enable MRTX1133 binding. The MRAS-R105H single mutant failed to bind MRTX1133, and the MRAS-F74Y/R105H double mutant showed only weak binding in the GTP-bound state. However, the triple mutant F74Y/R105H/L109Q exhibited high-nanomolar binding in both nucleotide states, and the quadruple mutant F74Y/R105H/F106Y/L109Q (MRASmut) bound MRTX1133 with low-nanomolar affinity in both GDP- and GMPPNP-bound forms (Fig. 7b and Supplementary Fig. 7b). Crystal structures of MRASmut bound to MRTX1133 in both

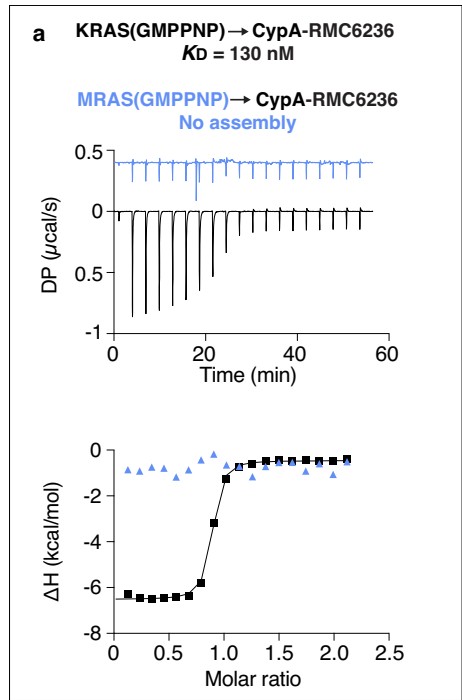

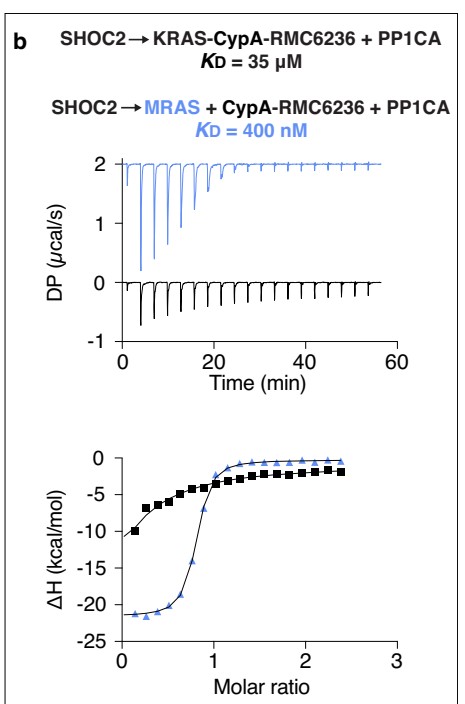

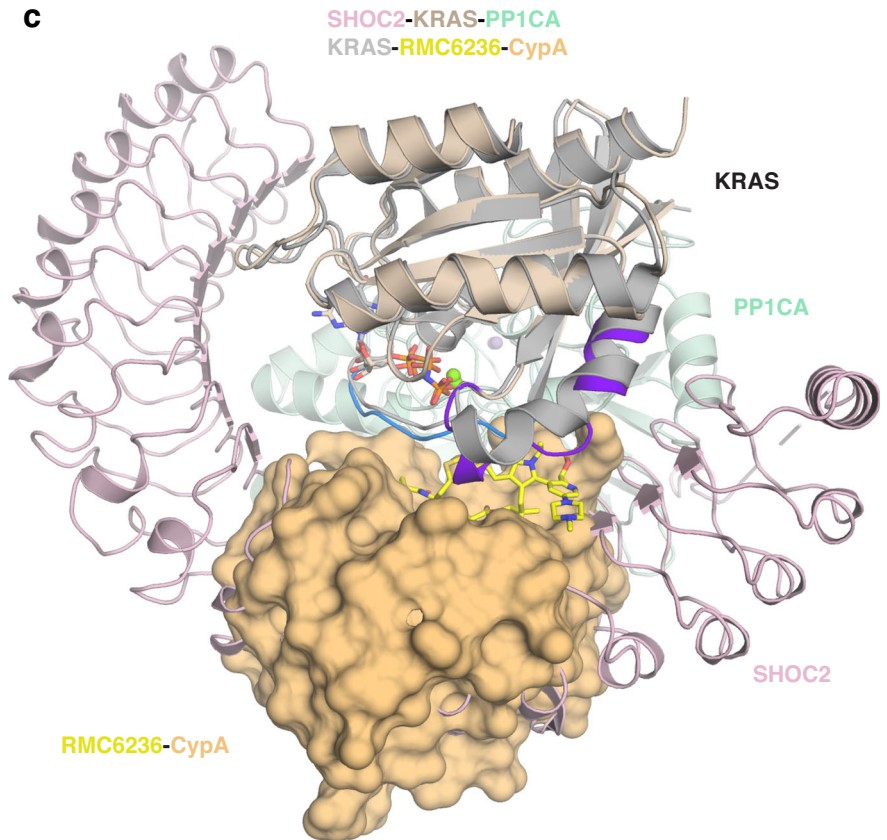

**Fig. 6 | Differential impact of RMC-6236−CypA on KRAS and MRAS complexes.**
**a** ITC traces showing the RMC-6236−CypA complex binding to KRAS (black) and MRAS (blue). The $K_D$ is calculated from two technical replicates. **b** ITC profiles showing the RMC6236·CypA complex weakens SKP formation (black) and has no effect on SMP formation (blue). The $K_D$ is calculated from two technical replicates. **c** Superposition of the SKP complex (light pink, brown, and light green, respectively, with Switch-I and -II shown in blue and purple, respectively) with the trimeric KRAS-RMC6236-CypA structure (PDB 9AX6 gray and light orange, respectively). CypA, shown as a surface, occupies the SHOC2-binding site.

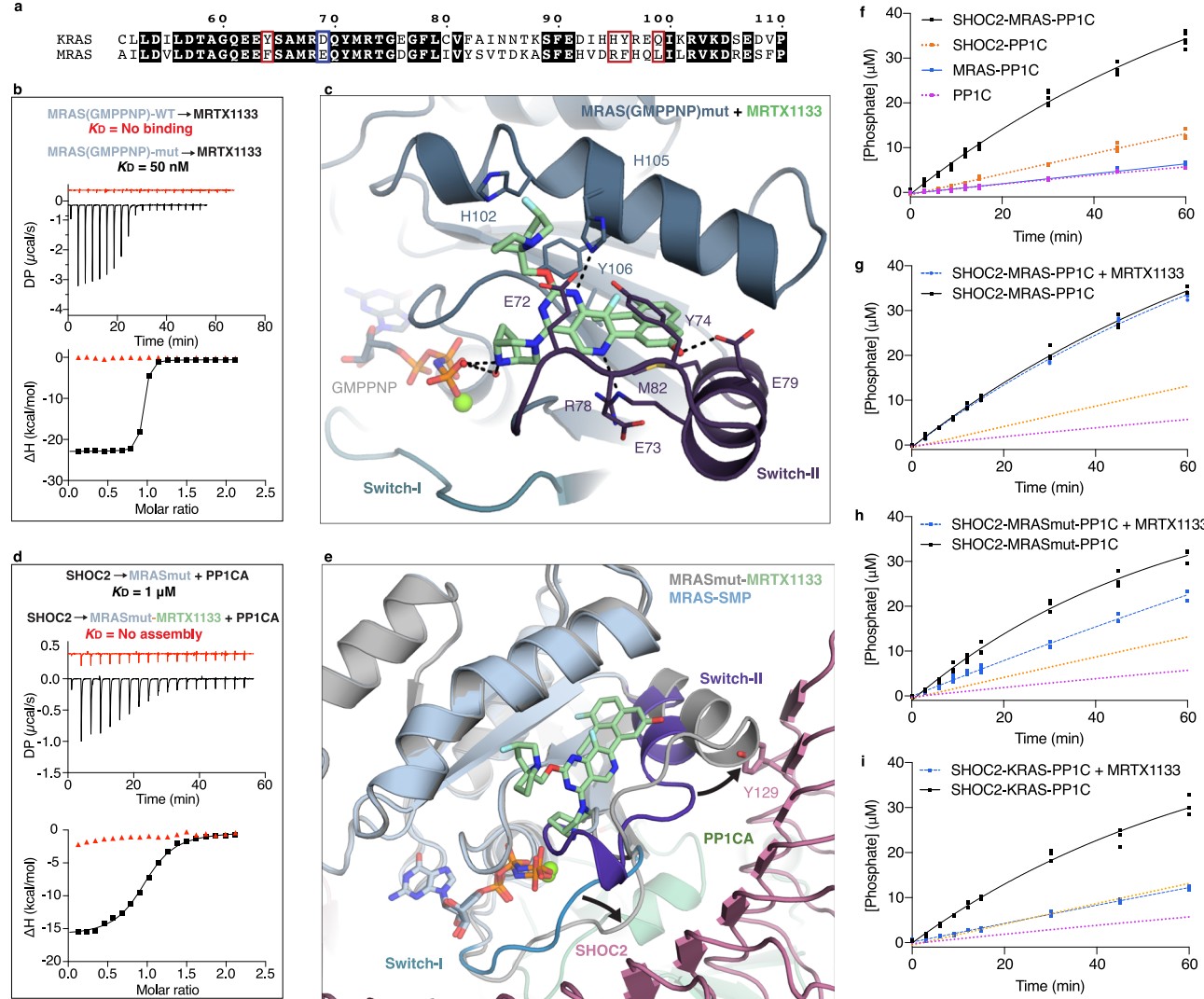

**Fig. 7 | Engineering MRTX1133 sensitivity into MRAS reveals inhibition of SHOC2·MRASmut·PP1C assembly by MRTX1133. a** Sequence alignment of KRAS and MRAS, highlighting residue differences within the MRTX1133 binding pocket. MRAS residues mutated in this study are boxed in red, while MRAS-E79, which forms part of the binding pocket but was not mutated, is boxed in blue. **b** ITC profile showing MRTX1133 binding to GMPPNP-bound wild-type MRAS (red) and engineered MRASmut (black). The $K_D$ is calculated from two technical replicates. **c** Crystal structure of the MRASmut(GMPPNP)·MRTX1133 complex, with MRASmut in dark blue and MRTX1133 in green. **d** ITC traces of SHOC2·MRASmut·PP1C formation in the absence (black) and presence of MRTX1133 (red). The $K_D$ is calculated from two technical replicates. **e** Superposition of the MRASmut(GMPPNP)·

MRTX1133 complex (gray) with MRAS from the SMP complex (light blue, PDB 7TVF). Black arrows indicate conformational changes in MRASmut switch regions upon MRTX1133 binding. Switch-II (dark purple) in the inhibitor-bound conformation clashes with SHOC2 residue Y129 (pink), explaining impaired complex formation. **f** Phosphate production by 1 nM PP1C by itself, or in the presence of 1 μM of MRAS, SHOC2, or MRAS and SHOC2 using CRAF CR2-pS peptide as a substrate. Phosphate production by the **g** SHOC2·MRAS·PP1C, **h** SHOC2·MRASmut·PP1C, and **i** SHOC2·KRAS·PP1C complexes, respectively, from a CRAF CR2-pS peptide as a substrate in the presence or absence of MRTX1133. Magenta and orange dashed lines represent the basal activity of PP1C and SHOC2·PP1C from (**f**), respectively.

nucleotide states were determined at 1.5 Å and 1.9 Å resolution and closely resembled the wild-type KRAS–MRTX1133 structures (Supplementary Table 3; Fig. 7c and Supplementary Fig. 8a–d).

Despite these mutations, MRASmut retained the ability to assemble the SMP complex with SHOC2 and PP1C, with affinity similar to wild-type MRAS (Fig. 7d). However, MRTX1133 successfully prevented SHOC2·MRASmut·PP1C complex formation, although it did not disassemble pre-formed complexes (Fig. 7d and Supplementary Fig. 8e). Structural modeling showed that MRTX1133 binding in MRASmut displaces the α2 helix and disrupts Switch-I engagement by SHOC2, mimicking the mechanism observed for SKP disruption (Fig. 7e). Although MRTX1133 prevented the assembly of SHOC2·MRASmut·PP1C and SHOC2·KRAS·PP1C complexes in biophysical assays, its effect on catalytic activity remained unclear. To address this, we used an in vitro dephosphorylation assay, which

monitors the release of phosphate from a CRAF CR2-pS peptide substrate. PP1C alone exhibited weak dephosphorylation activity (Fig. 7f), whereas SHOC2 and MRAS together synergistically activated PP1C activity to a significantly higher level. SHOC2 alone weakly stimulated PP1C, while MRAS alone did not stimulate PP1C (Fig. 7f). MRTX1133 had no effect on the catalytic activity of the SHOC2·MRAS·PP1C complex, as MRTX1133 cannot bind MRAS (Fig. 7g). The catalytic activity of the SHOC2·MRASmut·PP1C and SHOC2·KRAS·PP1C complexes was partially and completely inhibited by MRTX1133, respectively, consistent with its higher binding affinity for KRAS than for MRASmut (Fig. 7h, i). Furthermore, we observe that a RAS-less cell line devoid of HRAS, KRAS, NRAS, and MRAS transfected with MRASmut-Q71L treated with MRTX1133 could prevent dephosphorylation of CRAF, relative to MRAS-Q71L (Supplementary Fig. 9).

Recently, the Novartis team reported several compounds that bind to a pocket on SHOC2, thereby preventing RAS engagement with SHOC2 and PP1C[50]. These compounds block complex assembly by all RAS proteins, including MRAS. They reported a crystal structure of SHOC2 bound to Compound(R)-5 (Cmpd(R)-5), which has an affinity of ~1 μM for SHOC2[50]. In our dephosphorylation assay, we observe that Cpmd(R)-5 partially inhibits the activity of SHOC2-MRAS-PP1C and SHOC2-MRASmut-PP1C complexes and strongly inhibits the activity of SHOC2-KRAS-PP1C complex due to the weaker affinity of KRAS than MRAS for SHOC2 and PP1C (Supplementary Fig. 10).

Together, these findings establish a structural and functional basis for dual targeting of oncogenic KRAS and MRAS. As a proof-of-concept strategy, they demonstrate that inhibitors capable of blocking both SKP and SMP complex formation, either by engaging Switch-II directly, by stabilizing conformations incompatible with SHOC2 binding, or blocking SHOC2 binding, can effectively interfere with SHOC2-mediated signaling. Both MRTX1133 and RMC-6236 robustly inhibited SKP formation but only weakly dissociated existing ternary complexes. These results provide key mechanistic insights for therapeutic implications: preventing nascent complex assembly is comparatively more tractable than attempting to dislodge stable ternary complexes. The ability to prevent both SKP and SMP formation expands the scope of SHOC2-targeted RAS pathway inhibition, providing a framework for the rational development of dual-target inhibitors to overcome adaptive resistance in RAS-driven cancers.

## Discussion

In normal cells, the predominant RAS isoform (MRAS or canonical H/K/NRAS) that assembles with SHOC2 and PP1C for RAF activation remains unclear, due to redundancy among RAS proteins[14,45]. Our biophysical and structural data show that KRAS, the representative of the canonical RAS proteins, can also form a ternary complex with SHOC2 and PP1C, but with ~7-fold weaker affinity than MRAS. Although the stabilized SKP complex closely resembles SMP in overall architecture, wild-type KRAS lacks key MRAS-specific features: an N-terminal extension important for PP1CA binding, a longer β5-α4 loop that interacts with SHOC2, and interswitch residues that stabilize the complex[14]. These differences reduce the KRAS-SHOC2-PP1CA interface, explaining the lower affinity of SKP assembly. Thus, under normal physiological conditions, KRAS likely functions primarily to recruit RAF to the membrane via RBD-CRD interactions, whereas MRAS preferentially assembles the SHOC2-PP1C complex for RAF activation (Fig. 8a).

In oncogenic settings, however, mutants of KRAS (KRASmut), increase the GTP-bound population by impairing hydrolysis and/or enhancing nucleotide exchange, leading to elevated levels of active KRASmut. Active KRASmut, such as Q61 or G13 mutations that exhibit a strong dependency on SHOC2 as seen in CRISPR dependency data, promotes formation of SHOC2-KRASmut-PP1C complexes, driving MAPK activation through dephosphorylation (Fig. 1a)[15,16,21,50]. Despite KRAS binding weaker than MRAS, the high abundance of GTP-loaded KRASmut facilitates both RAF recruitment and SKP assembly, driving RAF dimerization and MAPK hyperactivation independently of upstream input (Fig. 8b). Given that MRAS knockout mice are viable and show no overt developmental defects[19], MRAS—and by extension the SMP complex—is not essential for RAF activation under physiological conditions, despite its higher binding affinity. In contrast, under oncogenic conditions, SKP assembly becomes critical for RAF dimerization and sustained MAPK hyperactivation, as SHOC2 is required for RAS-driven tumorigenesis and its loss suppresses RASmut-induced transformation and tumor growth in vivo[51,52].

Efforts to target oncogenic mutants of KRAS have historically been hindered by a lack of suitable binding pockets, though in the last few years, inhibitors targeting the induced-fit switch-II pocket have led to the approval of covalent KRAS-G12C inhibitors such as sotorasib and adagrasib for clinical use. However, resistance often arises through

secondary mutations, gene amplification, and MAPK pathway reactivation[8,27,42]. Notably, MRAS upregulation has been implicated in resistance via mislocalization of SCRIB from the SHOC2-SCRIB-PP1C complex, leading to Hippo pathway inactivation, nuclear YAP translocation, and increased MRAS expression (Fig. 8c). These findings suggest that MRAS-driven SMP complex formation may contribute to adaptive resistance, emphasizing the need to target both SKP and SMP complexes.

MRAS does not bind the KRAS-G12D inhibitor MRTX1133 due to sequence divergence within the Switch-II pocket, nor does it engage the pan-RAS inhibitor RMC-6236, likely because of differences in the Switch-I/II regions. The introduction of four Switch-II mutations into MRAS (MRASmut) conferred nanomolar affinity for MRTX1133 without disrupting its ability to form the SMP complex. However, MRTX1133 binding to MRASmut inhibited SMP complex assembly, providing proof-of-concept that pharmacological targeting of both KRAS and MRAS can disrupt SHOC2-RAS-PP1C complexes. These findings support a therapeutic strategy aimed at co-targeting KRAS and MRAS to suppress MAPK pathway reactivation in RAS-driven cancers (Fig. 8d).

Recently, MRAS has been reported to lack classical switch function as it fails to exchange GDP for GTP in vitro, even when membrane-tethered or exposed to SOS1 as measured by HPLC and in-cell NMR studies[53]. Conversely, several cellular studies have shown MRAS can adopt a GTP-bound state, albeit at low levels compared to HRAS, which can be weakly exchanged by GEFs such as RasGRF, CalDAG-GEFII, CalDAG-GEFIII, and SOS1[19,54,55]. In mouse embryonic fibroblasts lacking HRAS and NRAS, MRAS accumulate in their GTP-bound state after KRAS depletion, contributing to phospho-AKT signaling[56]. Overall, these results suggest that in vitro MRAS predominantly binds GDP due to a lack of intrinsic nucleotide exchange, though GEFs are able to stimulate nucleotide exchange to the GTP-bound state weakly[19,53–55]. However, there could be other unknown GEFs in vivo that may further elevate the cellular levels of MRAS-GTP[54,55,57,58], though a limited pool of MRAS-GTP would likely be sufficient to support catalytic amounts of SMP complex formation due to its high affinity.

Several drugs and small molecules have been developed to target oncogenic canonical RAS isoforms, yet no such inhibitors currently exist for MRAS, leaving the SMP complex untargeted. A pan-RAS inhibitor capable of engaging MRAS would block the assembly of the SMP complex and prevent the specific dephosphorylation of RAF. Our results demonstrate that MRAS can be engineered to bind the KRAS inhibitor MRTX1133 through a defined set of mutations, suggesting that the rational design of RAS or pan-RAS inhibitors such as MRTX1133 and RMC-6236 analogs could yield compounds with selective or dual affinity for both KRAS and MRAS. Since SHOC2 and PP1C are common to both SKP and SMP complexes, targeting either could disrupt complex formation broadly[21,33,59]. However, PP1C is unlikely to be a viable target due to the presence of multiple isoforms and its extensive interaction network with over 200 cellular proteins[13,21,33]. Our structure of the stabilized SKP complex, and its strong similarity to the SMP complex, suggests an alternative strategy: targeting SHOC2 at the conserved SHOC2-RAS interface, where Switch-I and -II of RAS engage, which has been recently demonstrated by Hauseman et al.[50]. A compound that disrupts this interaction could effectively inhibit the assembly of both SKP and SMP complexes and represents a promising direction for the development of SHOC2-RAS-PP1C-targeted therapies.

## Methods
### Cloning of expression constructs
The Entry clones for the single expression constructs PP1CA (2–300), PP1CA (7–300), MRAS (1–178), MRASmutants (1–178), KRAS (1–169), KRASmutant (1–169), and CypA (1–165) were synthesized (ATUM, Inc.) as codon-optimized fragments for Escherichia coli expression, downstream of a TEV protease site (ENLYFQ/G). Entry clones were

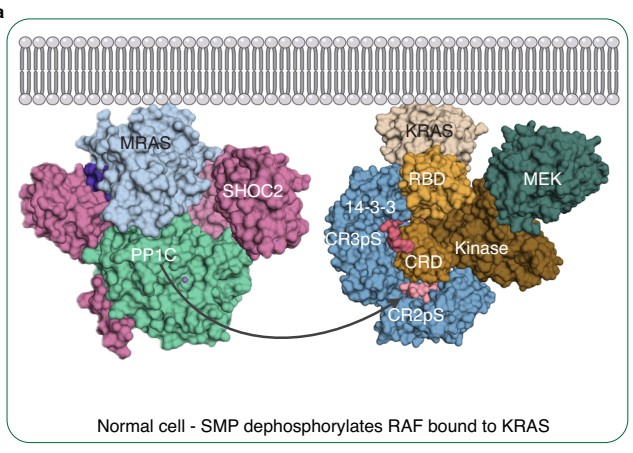

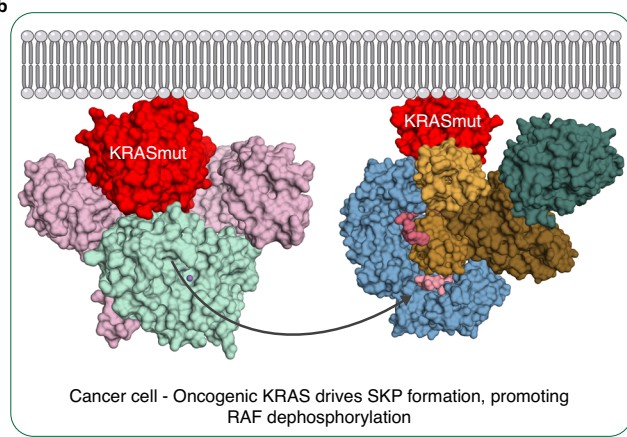

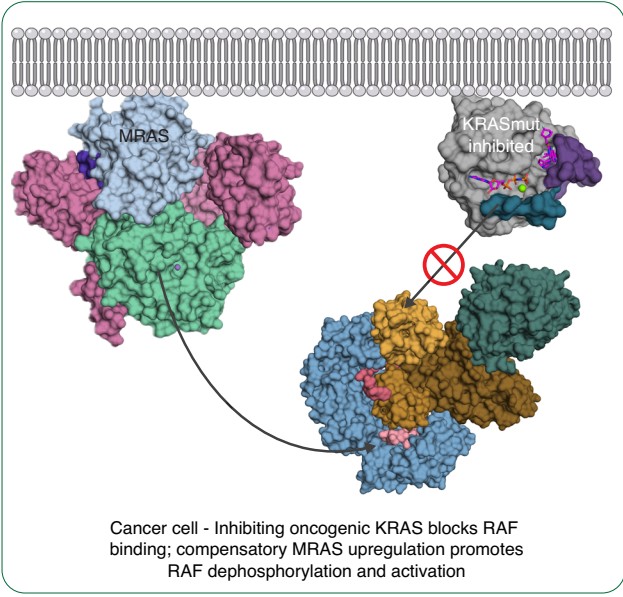

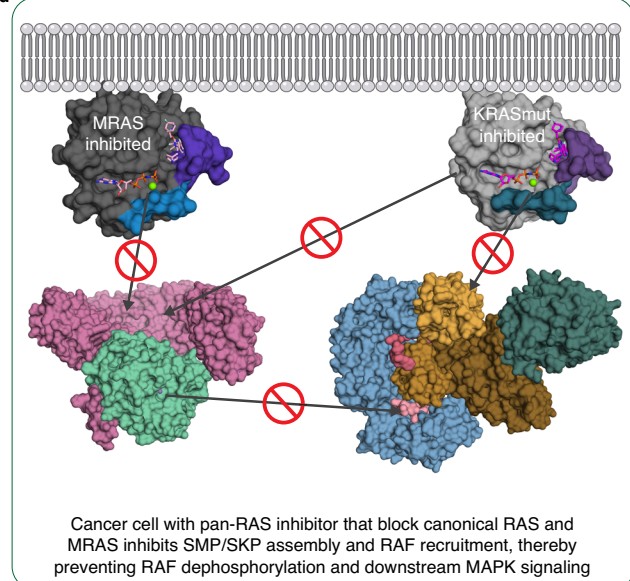

**Fig. 8 | Model of RAF regulation by oncogenic KRAS, MRAS, and pan-RAS inhibition. a** In normal cells, KRAS sequentially binds the RBD and RBD–CRD region of RAF, while MRAS assembles the SMP complex to dephosphorylate the CR2-pS site of RAF as part of its activation. **b** Oncogenic KRAS (KRASmut) interacts with RAF similarly and localizes it to the plasma membrane. Elevated GTP-bound levels of oncogenic KRAS also form the SKP complex, enabling CR2-pS

dephosphorylation. **c** Inhibition of oncogenic KRAS prevents localization of RAF at the membrane. Feedback activation of RAF occurs via MRAS upregulation, SMP complex formation, and continued CR2-pS dephosphorylation. **d** A pan-RAS inhibitor targeting both KRAS and MRAS would prevent SHOC2-RAS-PP1C complex assembly as well as RAF recruitment to the plasma membrane.

transferred to Expression clones containing an amino-terminal His6-tag (pDest-527, Addgene no. 1518) or an amino-terminal His6-Maltose Binding Protein (MBP) tag (pDest-566, Addgene no. 11517) by Gateway LR recombination.

The Entry clones for the two polycistronic plasmids, SHOC2-M173I (2–582)-KRAS-Q61R(1-169)-PP1CA-P50R(2-330) + SUGT1 and SHOC2(2-582) + SUGT1, were synthesized and codon optimized for insect expression (with the exception of SHOC2(2-582) which was PCR amplified from an MGC cDNA template plasmid (Horizon, MHS6278-20275921) with flanking attB1 and attB2 sites and an amino-terminal TEV protease cleavage site). The Entry clones were combined with pDest-623 (Addgene no. 161878) or pDest-624 (Addgene no. 161879) in a Gateway multisite LR recombination reaction to generate Expression plasmids[60]. The Bac-to-Bac system (Thermo Fisher Scientific) was used to generate bacmid DNA in strain DE95[61].

### Protein expression and purification

SHOC2 and stabilized SKP complex were expressed with the SUGT1 chaperone in insect cells as described before[13,17]. Briefly, $1.5 \times 10^6$ cells/ml of serum-free adapted Sf9 cells grown in SF900 III medium

(100 mL) were transfected with DNA–cellfectin II lipid complex (70 µL of bacmid DNA:250 µL cellfectin II:500 µL SF900 III medium). The culture was incubated for 120 h at 27 °C before the cell culture supernatant was isolated, and the baculovirus titer was determined by qPCR (TaqMan Gene Expression assay for the baculovirus GP64 protein). 1–2 L of Tni-FNL cells in SF900 III medium ($7 \times 10^5$ cells/ml) were grown for 24 h at 21 °C to allow doubling before baculovirus infection with a MOI of 3[62]. Cell cultures were grown for 72 h at 21 °C before collection and lysis (100 ml/L of cells) in buffer A (20 mM HEPES, pH 7.4, 300 mM sodium chloride, and 1 mM TCEP) using a Microfluidizer. Clarified lysates ($100,000 \times g$, 30 min at 4 °C) were filtered (0.45-µm high flow PES filter) and captured using a 5-ml Ni Sepharose High-Performance column (GE Healthcare) equilibrated in 20 mM HEPES, pH 7.4, 300 mM sodium chloride, 1 mM TCEP, and 35 mM imidazole on an NGC chromatography system (Bio-Rad). The column was washed for 5 column volumes (CVs) with 7% buffer B (20 mM HEPES, pH 7.4, 300 mM sodium chloride, 1 mM TCEP, 500 mM imidazole) before a gradient elution over 20 CVs to 100% buffer B. The elution peak was dialyzed into buffer A and digested with His-tagged TEV (ratio of 1:20 v/v protease: pooled protein) overnight at 4 °C. His-tagged TEV

and cleaved His-MBP were captured by a second Ni Sepharose column while cleaved proteins eluted in the flow-through and by running a shallow gradient of 0–10% buffer B over 10 CVs. Cleaved proteins were concentrated and purified further by size-exclusion chromatography using a 16/600 Superdex S200 column (Cytiva) equilibrated with 20 mM HEPES, pH 7.4, 150 mM sodium chloride, and 1 mM TCEP.

CypA, RAS proteins, and mutants were expressed in *E. coli* and purified as before[63]. Briefly, 1–2 L cells in Dynamite media were grown at 37 °C until an $OD_{600\,nm}$ of 6–8 was reached before induction with 0.5 mM IPTG. Cells were grown for a further 18–20 h at 16 °C. Cells were collected, lysed, and purified as described above for the SHOC2 and stabilized SKP complex, except for the RAS proteins. For the RAS proteins, the S200 equilibration buffer contained 5 mM magnesium chloride.

PP1CA was expressed and purified in *E. coli* as described above for RAS proteins, except the cells also harbored the GroEL-expression plasmid, pG-tf2 (Takara Bio USA), and expression was conducted at 10 °C. The lysis buffer was 20 mM Tris-HCl, pH 8.0, 700 mM sodium chloride, 10% w/v glycerol,1 mM manganese sulfate, 1 mM TCEP, and 0.5% w/v Triton X-100. Immobilized metal affinity chromatography was performed in this buffer without Triton X-100. PP1CA was purified in an identical way to SHOC2 or stabilized SKP, except a 5 mL MBPTrap HP column (Cytiva) was used to capture undigested fusion protein after TEV digestion, which failed to immobilize on the Ni Sepharose column. Furthermore, the S200 equilibration buffer was 20 mM Tris-HCl, pH 8.0, 500 mM sodium chloride, 1.0 mM manganese sulfate, and 1 mM TCEP.

## Nucleotide exchange of RAS GTPases

The bound GDP nucleotide in all recombinant RAS proteins was exchanged with a non-hydrolysable GTP analog, GMPPNP. 200 μL of ~15 mg/ml RAS protein was diluted with 720 μL of 40 mM Tris, pH 8.0, 200 mM ammonium sulfate, and 0.1 mM zinc chloride. 20 μL of 200 mM GMPPNP (tetralithium salt, Jena Bioscience) and 60 μL of alkaline phosphatase-agarose beads (Sigma) were added, and the reaction was mixed for 3 hours at room temperature. The beads were removed by centrifugation (1500 × *g* for 2 min) before the addition of 12 μL of 200 mM GMPPNP and 20 μL of 1 M magnesium chloride. The reaction was mixed overnight at 4 °C. Excess nucleotide was removed through desalting (PD-10 desalting column, Cytiva) equilibrated with 20 mM HEPES, 150 mM sodium chloride, 5 mM magnesium chloride, 1 mM TCEP, pH 7.4. Nucleotide analysis was performed to confirm the exchange of GDP to GMPPNP.

## Cryo-EM sample preparation and data acquisition

The purified SHOC2-KRAS-PP1CA complex was initially screened via negative-stain TEM on a FEI Tecnai T20 TEM, utilizing a uranyl formate stain to estimate concentration for blotting on graphene oxide before proceeding to Cryo-EM. For cryo-EM, 2.7 μl of 0.08 mg/mL of the SHOC2-KRAS-PP1CA complex was plunge frozen in liquid ethane cooled by liquid nitrogen onto Quantifoil R2/2 200 mesh gold grids (Quantifoil Micro Tools GmbH) covered with a monolayer of graphene oxide (Graphene Supermarket, NY); these grids were produced in-house by following a published protocol[64]. Grids were briefly exposed to UV/Ozone for 10 min using a Helios 500 UV Ozone Cleaner (UVFAB, CA) immediately before blotting. Vitrification was performed using a ThermoFisher Scientific Vitrobot Mark IV plunger with a chamber temperature of 4 °C and a chamber humidity of 95%. Grids were then transferred to the FEI Titan Krios (Thermo Fisher Scientific) TEM microscope at the National Cryo-EM Facility (NCEF) at the Frederick National Laboratory for Cancer Research operating at 300 kV with a Gatan K3 direct electron detector (Gatan Inc., Pleasanton, CA). Dose-fractionated movie stacks of 40 frames were collected in counting mode using the Latitude software (Gatan Inc., Pleasanton, CA) at a nominal magnification of 105,000, a defocus range of −0.75 to −2.25 μm, and a pixel size of 0.873 Å. Image shift was utilized as an imaging strategy using 4 images per hole with one focus position, with an exposure time of 2.29 s and a total dose of 52.3 $e^-/Å^2$. One dataset was collected, resulting in a total of 6860 micrographs.

## Cryo-EM data processing, model building, and analysis

Data processing was carried out in CryoSPARC[65]. A total of 6860 exposures were preprocessed with the Patch Motion Correction and Patch CTF Estimation jobs. A combination of manual picking and iterative rounds of Topaz training was performed using the wrapper included with CryoSPARC, and after particle inspection 518,550 particles were extracted using a box size of 256[66]. These particles were subjected to iterative rounds of 2D classification to remove junk particles, and 199,681 particles were used to generate an initial model via the Ab-Initio Reconstruction Job. After 1 round of homogeneous refinement followed by non-uniform refinement, we were able to obtain a 3 Å reconstruction. Resolution was estimated using the gold-standard Fourier shell correlation (GSFSC) of 0.143.

The volume was sharpened with CryoSPARC B-factor-based sharpening as well as DeepEMhancer v0.13 using its wideTarget training model[67]. SHOC2 and PP1CA from PDB 7TVF and KRAS from PDB 5UFE were combined, and rigid-body fit into the sharpened map using Chimera1.6[68]. The combined model was then subjected to flexible fitting with IMODFIT1.03[69]. Sections that did not have sufficient detail were removed, and the model was iteratively refined with Coot, Phenix, and ISOLDE1.4[70–72]. The local resolution map was calculated using CryoSPARC's Local Resolution Estimation job. Data collection and refinement statistics for the structure are shown in Supplementary Table 2. For the analysis of the anisotropy of the cryo-EM map, the 3DFSC validation server (https://3dfsc.salk.edu/)[73] was used to generate a histogram and directional FSC plot, and CryoSPARC was used to assess the orientation diagnostics.

## Crystallization of RAS proteins with inhibitors

Crystallization screening was conducted at 20 °C using the sitting-drop vapor diffusion method. RAS proteins were mixed with an equal volume of reservoir solution (200 nL:200 nL) using an ARI Crystal Phoenix Robot (Hudson Lab Automation).

*KRAS(1-169)GDP + MRTX1133*: 620 μL of 192 μM of KRAS was mixed with 20 μL of 7.42 mM MRTX1133 (final concentration of DMSO 3.1%) and incubated overnight at 4 °C before concentration to a final volume of 160 μL (14.4 mg/ml). The complex was then screened. Crystals grew in 25.5% w/v PEG 4000, 170 mM ammonium sulfate, 15% v/v glycerol (condition C6 of the Wizard III/IV screen, Molecular Dimensions). Crystals were cryoprotected with 20% v/v glycerol. A 1.56 Å dataset was collected on beamline 24-ID-C at the Advanced Photon Source (Argonne).

*KRAS(1-169)GMPPNP + MRTX1133*: 500 μL of 515 μM of exchanged KRAS was mixed with 35 μL of 7.42 mM MRTX1133 (final concentration of DMSO 6.5%, 9.5 mg/ml) and incubated overnight at 4 °C. The complex was then screened. Crystals grew in 2 M lithium sulfate, 50 mM sodium cacodylate, pH 6.0, 15 mM magnesium chloride, 5 mM spermidine (condition H6 of the Nucleix screen, Hampton Research). Crystals were cryoprotected with 2 M lithium sulfate. A 1.9 Å dataset was collected on beamline 24-ID-C at the Advanced Photon Source (Argonne).

*MRASmut(1-178)GDP + MRTX1133:* 1800 μL of 200 μM of MRASmut was mixed with 54 μL of 7.42 mM MRTX1133 (final concentration of DMSO 2.9%) and incubated overnight at 4 °C before concentration to a final volume of 280 μL (26.3 mg/ml). The complex was then screened. Crystals grew in 20% w/v PEG 3350, 200 mM ammonium fluoride (condition A3 of the PEG/Ion screen, Hampton Research). Crystals were cryoprotected with 20% v/v glycerol. A 1.6 Å dataset was collected on beamline iO3 at Diamond.

*MRASmut(1-178)GMPPNP + MRTX1133:* 650 μL of 192 μM of exchanged MRASmut was mixed with 20 μL of 7.42 mM MRTX1133

(final concentration of DMSO 3.0%) and incubated overnight at 4 °C before concentration to a final volume of 200 µL (12.0 mg/ml). The complex was then screened. Crystals grew in 1.6 M magnesium sulfate, 100 mM MES, pH 6.5 (condition E6 of the Top96 screen, Molecular Dimensions). Crystals were cryoprotected with 20% v/v glycerol. A 1.8 Å dataset was collected on beamline iO3 at Diamond.

## Structure determination

All crystallographic data were indexed and integrated with XDS, except for MRASmut(1-179)GDP + MRTX1133, which was indexed and integrated with DIALS[74,75]. All data were scaled, truncated, and converted to structural factors using Aimless[74,76]. Molecular replacement was performed with MOLREP using KRAS (1-169)GDP-G12D bound to MRTX1133 (PDB 7RPZ) for KRAS datasets and MRAS(1-178)GDP (PDB 1X1R) for MRASmut datasets[76]. Molecular replacement was performed with MOLREP using KRAS (1-169)GDP-G12D bound to MRTX1133 (PDB 7RPZ) for KRAS datasets and MRAS(1−178)GDP (PDB 1X1R) for MRASmut datasets[76]. The models were rebuilt with Coot[72]. Refinement was initially carried out by Refmac5, followed by Phenix.Refine[71,76,77]. Interactions, buried surface areas, and contacts were analyzed using Protein interfaces, surfaces, and assemblies (PISA) and PDBSum server at the European Bioinformatics Institute[35,36]. Figures were generated with PyMOL. All crystallographic and structural analysis software was supported by the SBGrid Consortium[78]. Data collection and refinement statistics for the structures are shown in Supplementary Table 3.

## Binding affinity measurements using isothermal titration calorimetry

Technical duplicate isothermal titration calorimetry measurements were performed on a MicroCal PEAQ-ITC instrument (Malvern Panalytical). MRTX1133 and RMC6236 (Chemgood) were dissolved in DMSO to a final concentration of 7.4 mM and 18 mM, respectively. Most experiments were conducted in 30 mM HEPES, 500 mM sodium chloride, 1 mM magnesium chloride, 0.5 mM TCEP, 0.1 mM manganese chloride, 5% v/v glycerol, pH 7.5 by dialysis of the proteins. The high salt and glycerol were required to keep PP1CA soluble in isolation and allow a direct comparison across datasets. Previous SPR measurements were made in 150 mM sodium chloride[13].

*RAS proteins binding to MRTX1133*: MRTX1133 was placed in the cell (50 µM), and RAS protein was in the syringe (500 µM). DMSO was added to a final concentration of 5% v/v.

*SMP and SKP formation*: PP1CA and RAS proteins were placed in the cell (20−50 µM) with SHOC2 at a 10-fold higher concentration in the syringe (200−500 µM).

*Inhibition of SMP and SKP formation by MRTX1133 or RMC6236-CypA*: PP1CA, RAS protein, and either MRTX1133 or RMC6236-CypA were placed in the cell (40 µM) with SHOC2 at a 10-fold higher concentration in the syringe (400 µM). DMSO was added to a final concentration of 5% v/v.

The remaining experiments were conducted in 20 mM HEPES, 150 mM sodium chloride, 2 mM magnesium chloride, 1 mM TCEP, pH 7.5.

*RMC6236 binding to CypA*: RMC6236 was placed in the cell (45 µM), and CypA protein was in the syringe (450 µM). DMSO was added to a final concentration of 5% v/v.

*RMC6236-CypA binding to RAS proteins*: A 1:1 complex of RMC6236-CypA was placed in the cell (50 µM) and RAS protein in the syringe (500 µM). DMSO was added to a final concentration of 5 % v/v.

*RMC6236-CypA and MRTX1133 disassembly of SKP and SMP complexes*: Prepared 1:1:1 of the SKP and SMP complexes from the individual proteins. These were dialyzed into the above buffer. SKP or SMP were placed in the cell (40−50 µM), and either RMC6236-CypA or MRXT1133 were placed in the syringe (400−500 µM). DMSO was added to a final concentration of 5% v/v.

All experiments were measured at 25 °C, with 19 injections (1 × 0.4 µL and 18 × 2.2 µL) with 175 s spacings and a stirring speed of 750 rpm. Data were analyzed using the MicroCal PEAQ-ITC analysis software (v1.41, Malvern Panalytical) and a "one set of sites" model. Data were plotted with Prism 10. A representative run of each experiment is shown in the figures with the average $K_D$ reported for technical duplicate experiments. All ITC parameters ($K_D$, $\Delta H$, and -$T\Delta S$) have been tabulated as averages with their ranges. Furthermore, previous important measurements from the literature have been recorded (Supplementary Table 1).

## Phosphatase assays

For peptide dephosphorylation assays, an 800 µl stock of 1 nM PP1CA, 1 µM SHOC2, and 1 µM RAS was prepared in 20 mM Tris, 150 mM sodium chloride, 1 mM magnesium chloride, 0.5 mM TCEP, 0.1 mM manganese chloride, pH 7.4, using stocks of 160 nM, 40 µM, and 40 µM, respectively, of each protein. The 40 µM RAS stock solution (MRAS, MRASmut or KRAS in the GMPPNP-bound state) contained 40 µM RAS protein with either 5% DMSO or 50 µM of either MRTX1133 or Compound(R)-5 in 5% DMSO. Compound(R)-5 was synthesized by Enamine. The SHOC2-RAS-PP1C complexes were incubated at room temperature for at least 15 min. Insolution™ Microcystin-LR *Microcystis aeruginosa* (Sigma, 475821), an inhibitor of PP1CA, was diluted to 10 µM in water, and 1 µl was aliquoted to each well of a 96 Black Well Assay Plate (Corning 3603) to inhibit the reaction at the various time points. 10 mg of CRAF-CR2pS peptide (Ac-SQRQRSTpSTPNVHMV, Biomatik) was dissolved in 100 mM Tris, pH 8.0 to a final concentration of 8 mM. The SHOC2-RAS-PP1C complex solution was added to 10 µl of peptide, and 80 µl of this solution was taken and added to a Microcystin-LR coated well at various time points (0, 3, 6, 9, 12, 15, 30, 45, and 60 min). Assays were performed in triplicate at room temperature. Duplicate phosphate standards in buffer were generated from 0 to 40 µM. Duplicate controls of buffer, water, 100 µM CRAF-CR2pS peptide, and Microcystin-LR were included. Free phosphate was measured using the Malachite Green assay kit from Sigma (MAK307) by the addition and mixing of 20 µl of working reagent to each well, incubation for 30 min followed by absorbance measurements at 620 nm on a FLUOstar Omega plate reader (BMG Labtech). Absorbances were converted to free phosphate using a standard curve generated from the phosphate standards. Dephosphorylation assays were also performed with the binary complex of SHOC2-PP1CA and MRAS-PP1CA, and PP1CA alone at the same concentrations as above. Graphs and fits were generated with Prism 10.

## Generation of quadruple knockout HRAS/NRAS/KRAS/MRAS 293 cells

"RASless" (HRAS/NRAS/KRAS triple knockout) 293 cells were transfected with sgRNAs to knock out (KO) MRAS (sgMRAS-1: GGAGCAATACATGCGCACGG, sgMRAS-2: GTCATTCCCGATGATCCTCG), following the same procedure[79]. Single-cell clones were validated via Western Blotting and sequencing of the sgRNA target region. DNA from cell pellets was extracted using the Qiagen Blood and Cell DNA extraction kit according to manufacturer's instructions. DNA was amplified using Pfu Ultra II polymerase and corresponding primers (Table 1).

## Transfection and co-immunoprecipitation

Cells were transiently transfected with pDest302-EF1-3X FLAG-MRAS Q71L and pDest-Myc-SHOC2 plasmids using JetOPTIMUS transfection reagent according to the manufacturer's protocol. 48 h later, cells were treated with 3 µM MRTX1133 or DMSO control for 4 h. Cells were washed with PBS and lysed in cold Lysis Buffer (50 mM Tris pH 8.0, 150 mM sodium chloride, 5 mM magnesium chloride, 1% Triton-X-100, 1 mM DTT, protease (Roche), and phosphatase inhibitors (Millipore-Sigma). After centrifuging at 27,000 × g for 10 min at 4 °C, the cleared

## Table 1 | Primers

| Target | Sequence |
| --- | --- |
| sgMRAS-1.for | GGGGACAAGTTTGTACAAAAAAGCAGGCTcttcgagcagccctagagag |
| sgMRAS-1.rev | GGGGACCACTTTGTACAAGAAAGCTGGtgctcacctgtctttgacgc |
| sgMRAS-2.for | GGGGACAAGTTTGTACAAAAAAGCAGGCTggctgtgctatgcctgagat |
| sgMRAS-2.rev | GGGGACCACTTTGTACAAGAAAGCTGGaactaaggggagcccttcaa |

Six confirmed KO clones, three from MRAS sgRNA-1 and three from sgRNA-2, were pooled for clonal heterogeneity.

## Table 2 | Antibodies used in this study

| Antibodies | Dilution | Source |
| --- | --- | --- |
| Myc-tag Rabbit polyclonal Ab | 1:2000 | Cell signaling technology cat.# 2272 |
| PP1 alpha Rabbit polyclonal Ab | 1:2000 | Upstate cat.# 06-221 |
| FLAG Rabbit polyclonal Ab | 1:4000 | Sigma-Aldrich Cat.# F7425 |
| CRAF Mouse mAb | 1:2000 | BD Biosciences Cat.# 610152 |
| P-S259 RAF1 Rabbit polyclonal Ab | 1:2000 | Cell Signaling Technology Cat.# 9421 |
| Anti-Rabbit IgG (H&L) Antibody DyLight 800 | 1:15,000 | Invitrogen Cat.# SA5-35571 |
| Anti-Mouse IgG (light chain) antibody Alexa Fluor 680 | 1:15,000 | Jackson ImmunoResearch Cat.# 115-625-174 |

supernatant was split between tubes containing either anti-FLAG agarose or protein A/G agarose and anti-CRAF antibody. An aliquot of lysate was saved with 4X NuPAGE LDS. After 1 h rotating at 4 °C, tubes were centrifuged briefly to pellet beads. Supernatant was aspirated and beads were washed with cold wash buffer (50 mM Tris pH 8.0, 150 mM sodium chloride, 5 mM magnesium chloride, 1% Triton-X-100) 3 times. 1.5X NuPAGE LDS was added to drained beads, and samples were heated at 70 °C for 10 min prior to SDS-PAGE. Gels were transferred to a nitrocellulose membrane, which was then blocked with 5% milk/TBS-T for 1 h. Primary antibodies (Table 2) were incubated with membranes overnight in 3% BSA/TBS-T. Membranes were washed 3 × 5 min with TBS-T and were incubated with anti-rabbit or anti-mouse secondary antibodies at room temperature for 1 h. Following three washes with TBS-T, membranes were scanned on a Li-COR Odyssey scanner.

### Reporting summary
Further information on research design is available in the Nature Portfolio Reporting Summary linked to this article.

## Data availability
The atomic coordinates and structure factors have been deposited in the Protein Data Bank and can be accessed using accession numbers 9O65, EMD-70159 (stabilized SKP complex), 9OON (KRAS(1-169)GDP with MRTX1133), 9OOO (KRAS (1-169)GMPPNP with MRTX1133), 9OOP (MRASmut(1-178)GDP with MRTX1133), and 9OOQ (MRASmut(1-178) GMPPNP with MRTX1133). The structures used as initial models for molecular replacement are available in the PDB under accession codes 1X1R (MRAS), 7RPZ (KRAS(G12D)GDP-MRTX1133), and 7TVF (SMP complex). Structures utilized for superpositions, and analysis can be found in the PDB using accession codes 1NVU (HRAS-SOS1), 6OB2 (KRAS + NF1), 6XI7 (KRAS-CRAF), 7LC1 (KRAS-Sin1), 7RPZ (KRAS(G12D) GDP-MRTX1133), 7T47 (KRAS(G12D)GMPPCP-MRTX1133), 7TVF (SMP complex), 8B69 (KRAS-Rgl2), 9AX6 (KRAS-RMC6236-CypA) and 9C15 (KRAS-PI3Kα). The source data underlying Figs. 1d, 5a, 5c, 6a–b, 7b, 7d and 7f–i and Supplementary Figs. 5a, 5g, 6a–b, 7b–c, 8e, 9 and 10 are provided as a Source Data file. Source data are provided with this paper.

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

## Acknowledgements

We acknowledge Dominic Esposito, William Gillette, John-Paul Denson, Matt Drew, Peter H. Frank, Natalie Granato-Guerrero, Brianna Higgins, Min Hong, Jenna Hull, Jennifer Mehalko, Simon Messing, Ashley Mitchell, Shelley Perkins, Ivy Poon, Nitya Ramakrishnan, Katie Geis, Mukul Sherekar, Troy Taylor, and Nicolas Wright for production of protein reagents used in this work. We thank Timothy Waybright for performing the nucleotide exchange analysis. CryoEM data were collected at the NCI National Cryo–EM Facility at the Frederick National Laboratory for Cancer Research, and we thank Tara Fox and Thomas Edwards for their help with the data collection. RASless cells were generated by Matthew J. Sale and Madeleine R. Allison. The authors also acknowledge the use of the Frederick Research Computing Environment (FRCE). This research used resources of the Advanced Photon Source, a U.S. Department of Energy (DOE) Office of Science User Facility operated for the DOE Office of Science by Argonne National Laboratory under Contract No. DE-AC02-06CH11357. This work is based upon research conducted at the NECAT beamlines, which are funded by the NIGMS/NIH (P30 GM124165). This research was supported by an agreement between the Advanced Photon Source and the Diamond Light Source, the U.K.'s national synchrotron science facility, located at the Harwell Science and Innovation Campus in Oxfordshire, where the work was performed under proposal AU34315-3. This project was funded in part with federal funds from the National Cancer Institute, National Institutes of Health (NIH) Contract 75N91019D00024. The content of this publication does not necessarily reflect the views or policies of the Department of Health and Human Services, and the mention of trade names, commercial products, or organizations does not imply endorsement by the US Government.

## Author contributions

D.A.B. and D.K.S. initiated the project. D.A.B. prepared the cryo-EM sample. L.I.F. and J.R.P. performed negative stain EM, cryo-EM grid preparation, data collection, processing, model building, refinement, and analysis. T.S. produced in-house graphene oxide grids. J.F. helped with the image processing. L.C.Y. and R.G. prepared and performed cellular experiments. D.A.B. conducted crystallography, structural analysis, and ITC experiments. V.E.W. and K.R.G. helped with the cloning and preparation of recombinant proteins. D.A.B., L.C.Y., D.V.N., F.M., and D.K.S. contributed to experimental design and data analysis. D.A.B., L.I.F., and D.K.S. wrote the manuscript with input from all authors.

## Funding

## Competing interests

F.M. is a consultant for Ideaya Biosciences, Kura Oncology, Leidos Biomedical Research, Pfizer, Daiichi Sankyo, Amgen, PMV Pharma, OPNA-IO, and Quanta Therapeutics. He has received research grants from Boehringer–Ingelheim, and is a consultant for and cofounder of BridgeBio Pharma. The remaining authors declare no competing interests.

## Additional information

Daniel A. Bonsor ®[1,5], Lorenzo I. Finci[1,5], Jacob R. Potter ®[1], Lucy C. Young[2], Vanessa E. Wall ®[1], Ruby Goldstein de Salazar[2], Katie R. Geis[1], Tyler Stephens[3], Joseph Finney[4], Dwight V. Nissley ®[1], Frank McCormick ®[1,2] & Dhirendra K. Simanshu ®[1] ✉

[1]NCI RAS Initiative, Cancer Research Technology Program, Frederick National Laboratory for Cancer Research, Frederick, MD, USA. [2]Helen Diller Family Comprehensive Cancer Center, University of California, San Francisco, San Francisco, CA, USA. [3]Vaccine Research Center Electron Microscopy Unit, Cancer Research Technology Program, Frederick National Laboratory for Cancer Research, Frederick, MD, USA. [4]National Cryo-EM Facility, Cancer Research Technology Program, Frederick National Laboratory for Cancer Research, Frederick, MD, USA. [5]These authors contributed equally: Daniel A. Bonsor, Lorenzo I. Finci. ✉e-mail: Dhirendra.Simanshu@nih.gov

