## [Transparent Peer Review file · Nature Communications]

Structure of SHOC2-KRAS-PP1C Complex Reveals RAS Isoform-Specific Determinants and Insights into Targeting Complex Assembly by RAS inhibitors

Corresponding Author: Dr Dhirendra Simanshu

Version 0:

Reviewer comments:

Reviewer #1

(Remarks to the Author)

Structure of SHOC2-KRAS-PP1C complex Reveals RAS isoform-Specific Determinants and Insights into Targeting complex Assembly by RAS Inhibitors

The SHOC2-MRAS-PP1C (SMP) complex is integral to RAF activation in the MAPK signaling pathway. It facilitates RAF dephosphorylation, promoting dimerization and RAF activation. While KRAS can form a ternary complex with SHOC2 and PP1C, its affinity is significantly lower than that of MRAS, making it difficult to obtain a complex structure. To address this, researchers introduced single-point mutations into each complex component to enhance the affinity of the SHOC2-KRAS-PP1C variant (SKP') complex, enabling successful cryo-EM reconstruction.

To explore therapeutic strategies targeting this complex, two RAS inhibitors were employed: MRTX1133, a KRAS G12D-specific inhibitor, and RMC-6236, a pan-RAS inhibitor. These inhibitors disrupted SKP' assembly without affecting SMP formation. Additionally, mutations in MRAS (MRASmut) were engineered to enhance its affinity for a KRAS G12D inhibitor. The resulting MRASmut complex assembly was impeded by these inhibitors, suggesting a potential approach to downregulate aberrant MAPK signaling upstream of RAF.

Results from these studies indicate that targeting GTPase interactions with the SHOC2-PP1 complex could serve as a novel mechanism to inhibit MAPK pathway activation. However, the novelty of this approach is tempered by a recent Nature publication focusing on targeting NRAS Q61R-SHOC2 interactions using novel SHOC2 inhibitors that prevent native complex formation. Putting this study in context with the current findings could provide a more comprehensive understanding of targeting RASopathies and RAS-driven cancers. Further validation of these results will be needed to confirm the therapeutic potential of disrupting SHOC2-GTPase interactions in clinical settings.

Comments:

1. The investigators should cite the published SHOC2/NRAS Q61R paper (Hauseman et al., Nature 2025), and put the published findings in context with the current study. Moreover, testing the recently identified SHOC2 inhibitor to examine inhibition of the native SKP, SKP' and SMP complex would elevate the impact of the work.
2. Crystallographic statistics are generally adequate, though the $I/\sigma I$ is borderline low in the high-resolution shells for KRAS(GMPPCP)MRTX1133 and MRASmut(GDP)MRTX1133.
3. A key objective of this work is to understand how structural differences between SKP and SMP complexes give rise to differences in binding affinity. While the structural details associated with SKP' is informative, a fairly rare KRAS Q61R mutant and SHOC2/PP1 with Noonan mutations is used. This non-native complex has a comparable binding affinity with SMP (150 nM vs 120 nM), so it is unclear how this structural comparison explains native differences between SKP and SMP.
4. The earlier reported binding affinity of SMP is mentioned by the investigators to be 120 nM, while the calculated K_d by ITC

in this paper (Fig. 1D left panel) is 900 nM. Some explanation is needed here.

5. All ITC measurements are stated to be averages of 2 runs, yet error analyses and type of replicates (biological vs technical) are not provided for any of these measurements.

- a. The reported ITC data lack error values, which are necessary for interpreting the presented K_d values.
- b. Given the large amount of ITC data, it would be helpful to tabulate these results and compare with earlier reported K_d values and associated references (relevant ternary complexes, WT KRAS, KRAS-G12D with MRTX1133, etc.)
- c. It is ambiguous in Figures 1D, 5C, 6B and 7D which component is being titrated and which components were premixed (cell). Clarification in the results section or Figure legends would be helpful.
- d. For ITC runs that are not productive, both “No assembly” and “No binding” are used to describe the results. Consistent terminology (binding for RAS inhibitor complexes vs assembly for the ternary complexes) would be helpful.
- e. Figure 1d – for the thermogram in the middle (SHOC2+KRAS+PP1CA). Has the heat of dilution has been subtracted in the Wiseman plot (plateau doesn't seem to be a zero)?
- f. Figure 5a – The lack of data points in the sigmoidal region makes K_d determination questionable.

6. In SI Fig 4a, the K_d results appear to contradict current literature. Why does MRTX1133 bind WT tighter than G12D and why does MRTX1133 prefer active-state G12D over inactive? In the main text (middle of p9 – it is stated that ITC determined sub-nM affinity for MRTX1133 and KRAS G12D) – please reference current literature for comparison and provide justification.

a. ITC for KRAS(GMPPCP)G12D in Figure 4a –The lack of data points in the sigmoidal region makes K_d determination questionable.

b. The SKP' complex was assembled using the KRAS Q61R mutant. Was the binding affinity of this mutant for MRTX1133 assessed?

7. Mutations in MRAS were introduced by making KRAS substitutions to enable MRTX1133 binding. While innovative, why was this approach not applied to RMC-6236?

8. The author mentioned that 6 residues of the SWII binding pocket are different in MRAS and KRAS, with mutations made rendering MRTX1133 sensitive to MRAS binding. It would be helpful to describe whether the residues mutated are involved in additional interactions in the SKP/SMP complex and how they differ between SKP' and SMP.

9. The authors claim proof of concept for dual drug targeting. If key residues within MRAS are mutated to make it look like KRAS, then it's expected that the MRASmut would bind the KRAS inhibitor. A stronger proof of concept would be doing SAR on the KRAS inhibitor to make it a MRAS binder. While this is likely beyond the scope of this manuscript, a discussion would heighten impact of the work.

10. Clarification is needed- bottom of p11. The investigators state that “Active KRASmut promotes formation of SHOC2-KRASmut-PP1C complexes...” all active KRAS mutants? A certain subset?

11. Page 18, “Refinement was initially...by Phenix. Refine” this sentence is repeated.

12. It is difficult to follow Figure 1e, as both surface and cartoon models are used. It would be helpful to either change the transparency or present the surface and cartoon model side by side.

13. It is difficult to follow Figure 3e, as similar color schemes are used for both SMP and SKP' complexes. The investigators should either change the color scheme or present a superposition of the individual proteins in the SMP and SKP' complex.

14. Figure 6a, 7b – the nucleotide state of KRAS and MRAS should be described.

15. Throughout the text, the authors use SKP to refer to the SKP' variant. It would be helpful to clarify the nomenclature to clearly distinguish between native SKP and SKP'.

16. In Figure 8d, the investigators provide an illustration to convey the concept that- ‘A pan-RAS inhibitor targeting both KRAS and MRAS would block RAF recruitment’. However, the figure does not clearly demonstrate that inhibiting MRAS and KRAS leads to the inhibition of RAF activation.

Reviewer #2

(Remarks to the Author)

Reviewer #3

(Remarks to the Author)

Reviewer #4

(Remarks to the Author)

The manuscript, "Structure of SHOC2–KRAS–PP1C Complex Reveals RAS Isoform-Specific Determinants and Insights into Targeting Complex Assembly by RAS Inhibitors," presents a 3.0 Å cryo-EM structure of the SHOC2–KRAS–PP1C (SKP) ternary complex, stabilized by pathogenic mutations associated with Noonan syndrome and cancer. The structure offers detailed modeling of SHOC2's LRRs, KRAS, and PP1C, and reveals isoform-specific interactions that account for the lower stability of the SKP complex relative to the higher-affinity SHOC2–MRAS–PP1C (SMP) complex.

The authors also explore the therapeutic potential of targeting complex assembly with RAS inhibitors. They demonstrate that MRTX1133 and RMC-6236 disrupt SKP formation while sparing the SMP complex. To test whether SMP formation could also be inhibited, they engineered a mutant MRAS (MRASmut) capable of binding MRTX1133 and showed that formation of the SHOC2–MRASmut–PP1C (SMutP) complex was blocked. This suggests a potential strategy for dual inhibition—targeting both SKP and SMP complexes—if MRAS-selective RAS inhibitor analogs can be developed, offering an avenue for a possible oncogenic treatment.

The findings described in this manuscript are clearly presented and the structural analyses is well done. The fact that the SKP complex looks quite similar to the corresponding complex with MRAS (SMP) might raise the question of just how impactful is the advance here, although there are some notable differences that seem to be significant. Additional strengths of the work include the authors' structure of the SKP complex using pathogenic mutations associated with Noonan syndrome and cancer offers a nice starting point to understand the conserved SHOC2-RAS interface that could be targeted for drug development. The differential impact of MRTX1133 and RMC-6236 on canonical versus MRAS-dependent complexes also offers potentially important implications toward combatting therapeutic resistance. Moreover, engineering an MRTX1133-sensitive MRAS (MRASmut) variant offers a proof-of-principle that possible analogs of RAS of pan-RAS inhibitors with selectivity for MRAS could target SMP complex formation and potentially lead an oncogenic treatment. However, a limitation of this study lies in the lack of physiological validation of the engineered MRASmut variant. While structural fidelity of the mutant-SMP complex is inferred from cryo-EM analysis between the SKP and SMP complexes, the functional competence of the mutant complex—particularly its ability to catalyze RAF CR2-pS dephosphorylation—was not explored. Given the centrality of this enzymatic function to SHOC2-RAS-PP1C signaling, biochemical assays and downstream signaling integrity, as demonstrated by proper de-phosphorylation activity within the mutant SMP complex, would represent important functional data to go along with the ability of MRTX1133 to bind the engineered MRASmut complex. Thus, the authors should address this issue by performing in vitro assays comparing the dephosphorylation of CR2-pS-containing RAF by wild-type versus mutant MRASmut complexes and Western blots from cell-based functional assays, showing phosphorylation readouts of RAF in MRAS-knockout cells reconstituted with either wild-type or MRASmut. These experiments would validate the mutant complex's functional equivalence to the wild-type SMP complex and support the biological significance of its observed inhibition by MRTX1133.

Minor comments:

1. A thorough proof-reading of the manuscript is needed. There are various errors, for example in the Title: "Structure of SHOC2–KRAS–PP1C complex Reveals RAS Isoform-Specific Determinants and Insights into Targeting Complex Assembly by RAS Inhibitors"; "complex" should be capitalized.
2. In the "Results", the sentence MRTX1133 successfully prevented SMmutP complex formation... "SMmutP" was not mentioned in the text, is this a typo?
3. In the "Discussion", the sentence "Active KRASmut promotes formation of SHOC2-KRASmut-PP1C complexes, driving MAPK activation through dephosphorylatio" needs a citation.

Reviewer #5

(Remarks to the Author)

Bonsor et al. present the structure determination and analysis of the SHOC2-KRAS-PP1CA (SKP) complex, complementing earlier results using SHOC2-MRAS-PP1CA (SMP) complexes. To stabilise the SKP assembly, known disease mutations were used. It is assumed that the resulting structures are still informative for these complexes in general, rather than specifically only for complexes carrying disease mutations, which appears reasonable but is not directly demonstrated. Due to the involvement of these signalling complexes in cancer, these new findings are important and may enable discovery of next-generation therapeutics.

The proof-or-principle experiment on SKP/SMP complex disruption using inhibitors presented in the manuscript is interesting, though new compounds would have to be developed to target un-mutated MRAS. It is, however, understandable that this exceeds the scope of the current work.

The paper is well written and reports interesting new findings that advance our understanding of the assembly of RAS-signalling complexes and pave the way for the discovery of future cancer therapeutics. As such, the paper is a possible candidate for publication in Nature Communications. However, the comments below will need to be addressed in order for the paper to adhere to the technical standards established in the field. Some minor comments and questions listed below should also be addressed.

Cryo-EM validation: The authors should add the following items to the manuscript to support the accuracy of cryo-EM structure determination:

- FSC curves between the model and the cryo-EM map (model vs. map FSC), evaluated at FSC = 0.5
- Analysis of the anisotropy of the cryo-EM map, e.g. using the 3D FSC validation server
- Figures showing the cryo-EM map and the refined model for important sections of the structure or where atomic interactions have been deduced (e.g. Fig. 2c-e; Fig. 3d-i) to support the accuracy of the modelling.

X-ray structures: The ligand densities are convincing, indicating that high-quality electron density maps have been obtained (as expected at the stated resolutions). However, Table 2 shows that the R_{free} values are virtually identical across the X-ray structures, even though one might have expected the higher-resolution structures to exhibit lower R_{free} values because the structures can be built more accurately. At the same time, the validation reports indicate somewhat high Ramachandran outlier statistics for some structures. Could the authors double check their full models to ensure that they are built as well as possible at the reported resolutions?

Minor comments and questions:

Page 4, bottom: Are the cellular concentrations of these proteins high enough to allow complex formation, given the affinities cited, or is it possible that H/K/NRAS only gain access to ternary complex formation in the presence of affinity-increasing mutations?

Page 7, top: Is the number of salt bridges and hydrogen bonds taken directly from PISA output or has this been re-evaluated visually using the molecular model? If the latter, what criteria were applied?

Page 18, near the bottom: Three sentences are present twice (referencing MOLREP, REFMAC refinement, and PISA).

Fig. 2a: Local resolution maps are typically coloured such that high resolution regions are blue and low-resolution regions are red (mirroring low/high b-factors from crystallography). The authors should also consider changing the colour palette used because the current spectrum (the default in Chimera) can cause difficulty for those with impaired colour vision (e.g. Cramer et al., Nat. Commun. 2020; <https://www.nature.com/articles/s41467-020-19160-7>).

Fig. 4: Switch I should be labelled in the figure panels (like Switch II).

Version 1:

Reviewer comments:

Reviewer #1

(Remarks to the Author)

The authors have addressed most concerns, as well as those raised by other reviewers. Substantial revisions have been made to the manuscript, including modifications to figures and text, with new data in response to reviewer requests. Major revisions include:

1. Inclusion of Compound 5 in the study, demonstrates its effect on SHOC2–RAS–PP1C complex activity toward a CRAF substrate via dephosphorylation assays.
2. Addition of the standard error of the mean values for all ITC data in the Table, along with the K_d values reported by others.
3. Addition of “titrant → cell” labels in all ITC figures.
4. Repetition of ITC experiments to obtain additional data points in the sigmoidal region.
5. Improved visualization through updated surface and cartoon models.
6. Execution of in vitro dephosphorylation assays (with or without MRTX1133) using SHOC2 and PP1CA with MRAS, MRASmut, or KRAS against a phosphopeptide of CRAF-CR2pS, with a new figure included.
7. Addition of a new figure showing Western blot analysis in RAS-less cells transfected with GTP-locked MRAS(Q71L) or MRASmut(Q71L).
8. Inclusion of FSC evaluation at 0.5 in the relevant figure.
9. Rebuilding of MRASmut(GDP)+MRTX1133 structural regions to address Ramachandran outliers, with updated Ramachandran statistics included in the Table for all rebuilt structures.

In addition to these major updates, the authors have implemented several minor textual and figure improvements throughout the manuscript. However, one point mentioned in the rebuttal that remains unaddressed in the main text is that the authors have not clarified whether the ITC replicates are biological or technical. This information should be added to the figure legends. Moreover, the justification for the large discrepancy in K_d values (previously 120 nM vs. 900 nM by ITC) remains unclear. The explanation provided does not fully resolve this issue. Additionally, the authors' statement that “all other instances refer to wild-type SKP” appears inconsistent, since the SKP complex structures reportedly include one mutation each in SHOC2, KRAS, and PP1C—thus, these do not represent wild-type SKP.

Aside from these minor concerns, the revised manuscript is significantly improved and otherwise ready for publication.

Reviewer #2

(Remarks to the Author)

Reviewer #3

(Remarks to the Author)

Reviewer #4

(Remarks to the Author)

I have carefully reviewed the revised manuscript and the rebuttal provided by the authors. I feel that the authors have made a good response to my original critiques with my main issue being that they perform in vitro de-phosphorylation assays, which the authors have done.

Reviewer #5

(Remarks to the Author)

Bonsor, Finci, et al. have revised their manuscript according to the comments of several anonymous reviewers. While the manuscript has been improved, additional changes will be needed before it can be considered for publication.

Main comments:

1) The authors have generally addressed the reviewers' comments in a suitable manner in their response to reviewers, but in some cases, the changes to the manuscript itself are rather minimal. This applies particularly to some valid issues raised by reviewer #1, e.g. points 3, 4.

2) Along the same lines: From the rebuttal: "All ITC measurements reported are technical replicates, and the standard error of the mean has been calculated for each measurement. This information has now been explicitly stated in the Materials and Methods." - I could not find the terms "technical replicate" or "standard error of the mean" anywhere in the manuscript text (the latter appears in a supplementary table). Could the authors specify where this is stated in the methods (maybe line numbers would help)?

3) The validation of the structure determined by cryo-EM requires further adjustments:

i) The local resolution estimation in Fig. 2a is uninformative - almost everything is blue (3 Å), nothing is red (7 Å). The range of resolutions displayed within the color ramp should probably be 3 Å to 5 Å, not 3 Å to 7 Å.

ii) Supplementary Table 2: The model resolution determined by model vs. map FSC at FSC = 0.143 is meaningless. This value should be removed. The value at FSC = 0.5 should be retained, as this is the correct threshold.

iii) The use of DeepEMhancer and other advanced post-processing programs is not a problem as such, but use of DeepEMhancer maps for structure refinement and validation (e.g. phenix.real_space_refine and model vs. map FSC) is not generally an accepted practice in the field. Can the authors confirm that a conventionally post-processed map was used for determination of the model vs. map FSC? If not, this should be changed and the updated curve/cutoff value reported.

iv) In the interest of transparency, the authors may want to consider disclosing the use of DeepEMhancer in the figure legends where the maps treated with this algorithm are shown, rather than just in Supplementary Table 2 and the methods section.

Other minor comments:

The paper on 3D FSC validation (Tan et al., Nature Methods, 2017) should be cited.

Supplementary Fig. 10: Showing the compound in question (compound 5) would be helpful to the non-specialist reader.

Point-by-point response to reviewer's comments:

We sincerely appreciate the reviewers' thorough and constructive feedback. In this revised manuscript, we have addressed all concerns raised, resulting in an improved and more robust version. Below, we provide point-by-point responses to the reviewers' comments (highlighted in blue).

Reviewer #1:

Structure of SHOC2-KRAS-PP1C complex Reveals RAS isoform-Specific Determinants and Insights into Targeting complex Assembly by RAS Inhibitors

The SHOC2-MRAS-PP1C (SMP) complex is integral to RAF activation in the MAPK signaling pathway. It facilitates RAF dephosphorylation, promoting dimerization and RAF activation. While KRAS can form a ternary complex with SHOC2 and PP1C, its affinity is significantly lower than that of MRAS, making it difficult to obtain a complex structure. To address this, researchers introduced single-point mutations into each complex component to enhance the affinity of the SHOC2-KRAS-PP1C variant (SKP') complex, enabling successful cryo-EM reconstruction.

To explore therapeutic strategies targeting this complex, two RAS inhibitors were employed: MRTX1133, a KRAS G12D-specific inhibitor, and RMC-6236, a pan-RAS inhibitor. These inhibitors disrupted SKP' assembly without affecting SMP formation. Additionally, mutations in MRAS (MRASmut) were engineered to enhance its affinity for a KRAS G12D inhibitor. The resulting MRASmut complex assembly was impeded by these inhibitors, suggesting a potential approach to downregulate aberrant MAPK signaling upstream of RAF.

Results from these studies indicate that targeting GTPase interactions with the SHOC2-PP1 complex could serve as a novel mechanism to inhibit MAPK pathway activation. However, the novelty of this approach is tempered by a recent Nature publication focusing on targeting NRAS Q61R-SHOC2 interactions using novel SHOC2 inhibitors that prevent native complex formation. Putting this study in context with the current findings could provide a more comprehensive understanding of targeting RASopathies and RAS-driven cancers. Further validation of these results will be needed to confirm the therapeutic potential of disrupting SHOC2-GTPase interactions in clinical settings.

Comments:

1. The investigators should cite the published SHOC2/NRAS Q61R paper (Hauseman et al., Nature 2025) and put the published findings in context with the current study. Moreover, testing the recently identified SHOC2 inhibitor to examine inhibition of the native SKP, SKP' and SMP complex would elevate the impact of the work.

The publication by Hauseman *et al.* (Nature, 2025) on SHOC2/NRAS Q61R appeared in the public domain during the review process. The structures of SHOC2 bound to a cyclic peptide and Compound 5 were released by the PDB approximately one month before their publication.

We had ordered Compound 5 prior to the paper's release, without realizing that the highest-affinity SHOC2 binder described in the manuscript is Compound 6. While Compound 6 can be synthesized commercially, its lead time is more than 12 weeks, and thus, we proceeded with testing Compound 5 to avoid significant experimental delays.

In the revised manuscript, we have now cited this work. Due to its limited solubility (as noted in Hauseman *et al.*), we were unable to assess its effect on SHOC2-RAS-PP1C assembly using ITC. However, we demonstrate that Compound 5 affects the activity of SHOC2-RAS-PP1C complexes on a CRAF substrate via dephosphorylation assays. These data have now been included (Supplementary Fig. 10), alongside the effects of MRTX1133 on the activity of SHOC2-RAS-PP1C complexes, as requested by reviewer #2.

2. Crystallographic statistics are generally adequate, though the I/σ is borderline low in the high-resolution shells for KRAS(GMPPCP)MRTX1133 and MRASmut(GDP)MRTX1133.

We thank the reviewer for this comment. The final resolution was selected based on both $CC1/2$ and I/σ from the Aimless summary, rather than relying solely on I/σ .

3. A key objective of this work is to understand how structural differences between SKP and SMP complexes give rise to differences in binding affinity. While the structural details associated with SKP' is informative, a fairly rare KRAS Q61R mutant and SHOC2/PP1 with Noonen mutations is used. This non-native complex has a comparable binding affinity with SMP (150 nM vs 120 nM), so it is unclear how this structural comparison explains native differences between SKP and SMP.

The stabilization of the SKP through three pathogenic mutations does result in an affinity similar to WT SMP. However, our structural analysis reveals several differences in the SKP' interface compared to SMP, including loss of hydrogen bonds and van der Waal interactions. As described in the main text, only one of the mutations, PP1C-P50R, forms direct de novo contacts at an interface, while the others do so indirectly but still locally e.g. SHOC2-M173I is found at the interface, no direct contacts form with KRAS and the mutation does not affect distal residues in SHOC2, demonstrating that the mutations stabilize the interaction locally, while the remainder of the interface remains largely unchanged from native SKP. Although the KRAS Q61R mutant is relatively rare, the structure of the SMP complex by Hauseman *et al* (Nature, 2022) was determined using this stabilization mutation (as well as SHOC2-M173I) to increase complex affinity, which motivated its use in our study. Comparisons of the SMP complexes across all four research groups show practically identical interfaces, except for local conformational changes at the mutation site, further highlighting that stabilization does not alter the native interface.

4. The earlier reported binding affinity of SMP is mentioned by the investigators to be 120 nM,

while the calculated K_d by ITC in this paper (Fig. 1D left panel) is 900 nM. Some explanation is needed here.

The reported K_d values were obtained using two different techniques—SPR in the earlier study (120 nM) and ITC in this work (900 nM). Differences in immobilization, mass transport, and detection principles between SPR and ITC often lead to variations in absolute affinities. Furthermore, ITC required higher concentrations of salt to keep PP1C soluble at the concentrations required; however, the relative rank order of affinities across complexes is preserved in both cases.

5. All ITC measurements are stated to be averages of 2 runs, yet error analyses and type of replicates (biological vs technical) are not provided for any of these measurements.

All ITC measurements reported are technical replicates, and the standard error of the mean has been calculated for each measurement. This information has now been explicitly stated in the Materials and Methods.

a. The reported ITC data lack error values, which are necessary for interpreting the presented K_d values.

The standard error of the mean has been calculated, and the mean values are reported in the text. A summary of all the ITC data, along with their associated errors, has now been included in Supplementary Table 1.

b. Given the large amount of ITC data, it would be helpful to tabulate these results and compare with earlier reported K_d values and associated references (relevant ternary complexes, WT KRAS, KRAS-G12D with MRTX1133, etc.)

We agree with the reviewer and have now tabulated all our ITC data (Supplementary Table 1), alongside previously published measurements of binary and ternary protein interactions, as well as MRTX1133 and RMC6236 binding of KRAS and CypA proteins, respectively.

c. It is ambiguous in Figures 1D, 5C, 6B and 7D which component is being titrated, and which components were premixed (cell). Clarification in the results section or Figure legends would be helpful.

We thank the reviewer for this helpful suggestion. All ITC traces now show the titrant being injected into the cell as “titrant→cell” in their titles.

d. For ITC runs that are not productive, both “No assembly” and “No binding” are used to describe the results. Consistent terminology (binding for RAS inhibitor complexes vs assembly for the ternary complexes) would be helpful.

We have revised the figures to show “No binding” for RAS inhibitor complexes and “No assembly” for ternary complexes.

e. Figure 1d – for the thermogram in the middle (SHOC2+KRAS+PP1CA). Has the heat of dilution has been subtracted in the Wiseman plot (plateau doesn't seem to be a zero)?

The reviewer is correct that the heat of dilution had not been subtracted; this has now been corrected.

f. Figure 5a – The lack of data points in the sigmoidal region makes K_d determination questionable.

We agree with the reviewer. This data has been recollected with more injections of a smaller volume to accurately measure the thermodynamics of binding.

6. In SI Fig 4a, the K_d results appear to contradict current literature. Why does MRTX1133 bind WT tighter than G12D and why does MRTX1133 prefer active-state G12D over inactive? In the main text (middle of p9 – it is stated that ITC determined sub-nM affinity for MRTX1133 and KRAS G12D) – please reference current literature for comparison and provide justification.

We appreciate the reviewer's comment. As suggested, the ITC data have been recollected with more injections of a smaller volume to accurately measure the thermodynamics of binding. As stated in the methods, the ITC data were collected in 500 mM sodium chloride and 5% glycerol to facilitate comparisons when MRTX1133 is used in conjunction with SKP and SMP complexes, as PP1C precipitates at the concentrations required in the ITC at low salt concentrations. KRAS G12D binds MRTX1133 slightly weaker than WT under these conditions, most likely due to the higher concentration of salt interfering with the aspartic group forming hydrogen bonds to MRTX1133. With repeated and more accurate measurements, the GMPPNP-bound G12D mutant now binds slightly weaker than the GDP-bound G12D mutant. Minor discrepancies in K_D values with previously published values may also reflect the different sensitivity ranges of biophysical methods, such as SPR and ITC, among others. We have now cited relevant literature for comparison and provided justification for the observed differences in the main text and Supplementary Table 1.

a. ITC for KRAS(GMPPCP)G12D in Figure 4a –The lack of data points in the sigmoidal region makes K_d determination questionable.

We agree with the reviewer. This data has been recollected with more injections of a smaller volume to accurately measure the thermodynamics of binding.

b. The SKP' complex was assembled using the KRAS Q61R mutant. Was the binding affinity of this mutant for MRTX1133 assessed?

We have not measured MRTX1133 binding to KRAS Q61R by ITC, as the KRAS Q61R mutant was not used in any experiments related to MRTX1133. Recently, in a separate study (PMID:

40473215), KRAS Q61R was assessed by SPR and found to bind MRTX1133 with reduced affinity compared to KRAS WT.

7. Mutations in MRAS were introduced by making KRAS substitutions to enable MRTX1133 binding. While innovative, why was this approach not applied to RMC-6236?

We selected one of the two inhibitors for this proof-of-concept study. Considering that MRTX1133 is well characterized and a direct binder of RAS GTPase, we chose it for this study. Furthermore, MRTX1133 was easier to work with as we only needed to mutate the Switch-II pocket. To conduct RMC6236 measurements we would need to mutate both Switch-I, to allow the CypA-MRAS interface to form, and Switch-II to allow RMC6236 interaction to occur.

8. The author mentioned that 6 residues of the SWII binding pocket are different in MRAS and KRAS, with mutations made rendering MRTX1133 sensitive to MRAS binding. It would be helpful to describe whether the residues mutated are involved in additional interactions in the SKP/SMP complex and how they differ between SKP' and SMP.

Only one residue, Y64 (KRAS) or F74 (MRAS), in the SWII pocket interacts with both MRTX1133 and SHOC2 in the SMP or SKP complexes. These interactions occur through van der Waals interactions and have been noted in the main text.

9. The authors claim proof of concept for dual drug targeting. If key residues within MRAS are mutated to make it look like KRAS, then it's expected that the MRASmut would bind the KRAS inhibitor. A stronger proof of concept would be doing SAR on the KRAS inhibitor to make it a MRAS binder. While this is likely beyond the scope of this manuscript, a discussion would heighten impact of the work.

We would like to clarify that this point is addressed in the Results section, where we explain why MRAS was mutated:- “While ligand redesign could be explored to accommodate MRAS, we pursued a simpler approach: introducing mutations into MRAS Switch-II pocket to restore MRTX1133 binding.” We also highlight in the Discussion that “rational design of RAS or pan-RAS inhibitors, such as MRTX1133 or RMC-6236 analogs, could yield compounds with selective or dual affinity for both KRAS and MRAS,” acknowledging the potential for future SAR studies to extend this proof-of-concept.

10. Clarification is needed- bottom of p11. The investigators state that “Active KRASmut promotes formation of SHOC2-KRASmut-PP1C complexes...” all active KRAS mutants? A certain subset?

We thank the reviewer for this correction. We have rewritten this sentence and the next to state into a single sentence: “Active KRASmut, such as Q61 or G13 mutations that exhibit a strong dependency on SHOC2 as seen in CRISPR dependency data, promotes formation of SHOC2-KRASmut-PP1C complexes, driving MAPK activation through dephosphorylation.”

11. Page 18, “Refinement was initially...by Phenix. Refine” this sentence is repeated.

We thank the reviewer for their careful reading of the material and methods. The duplicated sentence has been removed.

12. It is difficult to follow Figure 1e, as both surface and cartoon models are used. It would be helpful to either change the transparency or present the surface and cartoon model side by side.

We thank the reviewer for their suggestion and have increased the transparency of the map to allow better visualization of the cartoon model.

13. It is difficult to follow Figure 3e, as similar color schemes are used for both SMP and SKP' complexes. The investigators should either change the color scheme or present a superposition of the individual proteins in the SMP and SKP' complex.

We combined Figures 3d and 3e into a single, larger figure that clearly shows the differences in Switch engagement of SHOC2 in SMP and SKP complexes.

14. Figure 6a, 7b – the nucleotide state of KRAS and MRAS should be described.

The nucleotide state of KRAS and MRAS is now described in the titles.

15. Throughout the text, the authors use SKP to refer to the SKP' variant. It would be helpful to clarify the nomenclature to clearly distinguish between native SKP and SKP'.

We would like to keep the nomenclature as is because SKP' is only used in a single ITC experiment to measure affinity and in our structural determination; all other instances refer to wild-type SKP. We note that previously reported SMP complex structures by four different groups contained mutations in MRAS and truncations in SHOC2, yet were referred to as SMP complexes throughout. For consistency, we would prefer to retain the name SKP complex, and we have clearly stated at the outset that SKP complex structures include one mutation each in SHOC2, KRAS, and PP1C.

16. In Figure 8d, the investigators provide an illustration to convey the concept that- 'A pan-RAS inhibitor targeting both KRAS and MRAS would block RAF recruitment'. However, the figure does not clearly demonstrate that inhibiting MRAS and KRAS leads to the inhibition of RAF activation.

We thank the reviewer for their careful review of our figures. We agree that the figure did not explicitly convey RAF recruitment, and we have now updated it to clearly indicate that inhibition of both KRAS and MRAS blocks RAF recruitment and activation.

Reviewer #2:

The manuscript, “Structure of SHOC2–KRAS–PP1C Complex Reveals RAS Isoform-Specific Determinants and Insights into Targeting Complex Assembly by RAS Inhibitors,” presents a 3.0 Å cryo-EM structure of the SHOC2–KRAS–PP1C (SKP) ternary complex, stabilized by pathogenic mutations associated with Noonan syndrome and cancer. The structure offers detailed modeling of SHOC2’s LRRs, KRAS, and PP1C, and reveals isoform-specific interactions that account for the lower stability of the SKP complex relative to the higher-affinity SHOC2–MRAS–PP1C (SMP) complex.

The authors also explore the therapeutic potential of targeting complex assembly with RAS inhibitors. They demonstrate that MRTX1133 and RMC-6236 disrupt SKP formation while sparing the SMP complex. To test whether SMP formation could also be inhibited, they engineered a mutant MRAS (MRASmut) capable of binding MRTX1133 and showed that formation of the SHOC2–MRASmut–PP1C (SMutP) complex was blocked. This suggests a potential strategy for dual inhibition—targeting both SKP and SMP complexes—if MRAS-selective RAS inhibitor analogs can be developed, offering an avenue for a possible oncogenic treatment.

The findings described in this manuscript are clearly presented and the structural analyses is well done. The fact that the SKP complex looks quite similar to the corresponding complex with MRAS (SMP) might raise the question of just how impactful is the advance here, although there are some notable differences that seem to be significant. Additional strengths of the work include the authors’ structure of the SKP complex using pathogenic mutations associated with Noonan syndrome and cancer offers a nice starting point to understand the conserved SHOC2-RAS interface that could be targeted for drug development. The differential impact of MRTX1133 and RMC-6236 on canonical versus MRAS-dependent complexes also offers potentially important implications toward combatting therapeutic resistance. Moreover, engineering an MRTX1133-sensitive MRAS (MRASmut) variant offers a proof-of-principle that possible analogs of RAS of pan-RAS inhibitors with selectivity for MRAS could target SMP complex formation and potentially lead an oncogenic treatment.

However, a limitation of this study lies in the lack of physiological validation of the engineered MRASmut variant. While structural fidelity of the mutant-SMP complex is inferred from cryo-EM analysis between the SKP and SMP complexes, the functional competence of the mutant complex—particularly its ability to catalyze RAF CR2-pS dephosphorylation—was not explored. Given the centrality of this enzymatic function to SHOC2-RAS-PP1C signaling, biochemical assays and downstream signaling integrity, as demonstrated by proper dephosphorylation activity within the mutant SMP complex, would represent important functional data to go along with the ability of MRTX1133 to bind the engineered MRASmut complex. Thus, the authors should address this issue by performing in vitro assays comparing the dephosphorylation of CR2-pS-containing RAF by wild-type versus mutant MRASmut complexes and Western blots from cell-based functional assays, showing phosphorylation

readouts of RAF in MRAS-knockout cells reconstituted with either wild-type or MRASmut. These experiments would validate the mutant complex's functional equivalence to the wild-type SMP complex and support the biological significance of its observed inhibition by MRTX1133. We thank the reviewer for this suggestion. To address this, we performed in vitro dephosphorylation assays using SHOC2 and PP1CA with either MRAS, MRASmut or KRAS against a phosphopeptide of CRAF-CR2pS. All three RAS proteins (MRAS, MRASmut, and KRAS) synergistically stimulated PP1CA activity in the presence of SHOC2, whereas SHOC2 alone only weakly stimulated activity, and RAS alone had no effect (Figure 7f).

We also performed these assays in the presence of MRTX1133. As expected, MRTX1133 had no effect on SMP activity because it does not bind MRAS (Figure 7g). Consistent with previous observations, MRTX1133 inhibits SKP activity and partially inhibits SHOC2-MRASmut-PP1C complex activity, likely reflecting the higher affinity of the SHOC2-MRASmut-PP1C complex compared to SKP (Figure 7h and 7i).

We also performed Western blots in RAS-less cells (lacking HRAS, KRAS, NRAS, and MRAS), transfected with MRAS or MRASmut, both carrying the Q71L mutation to maintain the GTP-bound state. MRTX1133 had no effect on RAF dephosphorylation in MRAS-expressing cells, whereas a partial reduction was observed in MRASmut cells (Supplementary Figure 9).

Minor comments:

1. A thorough proof-reading of the manuscript is needed. There are various errors, for example in the Title: “Structure of SHOC2–KRAS–PP1C complex Reveals RAS Isoform-Specific Determinants and Insights into Targeting Complex Assembly by RAS Inhibitors”; “complex” should be capitalized.

We thank the reviewer for their careful reading of our manuscript. We have capitalized “complex” in the title and have thoroughly reviewed the manuscript to correct any other errors.

2. In the “Results”, the sentence MRTX1133 successfully prevented SMmutP complex formation... “SMmutP” was not mentioned in the text, is this a typo?

We thank the reviewer for pointing this out. This was a typo, and “SMmutP” to “SHOC2-MRASmut-PP1C complex”.

3. In the “Discussion”, the sentence “Active KRASmut promotes formation of SHOC2-KRASmut-PP1C complexes, driving MAPK activation through dephosphorylation” needs a citation.

We have added the Hauseman *et al* (Nature, 2025) as a citation to support this statement.

Reviewer #5

Bonsor et al. present the structure determination and analysis of the SHOC2-KRAS-PP1CA (SKP) complex, complementing earlier results using SHOC2-MRAS-PP1CA (SMP) complexes. To stabilise the SKP assembly, known disease mutations were used. It is assumed that the resulting structures are still informative for these complexes in general, rather than specifically only for complexes carrying disease mutations, which appears reasonable but is not directly demonstrated. Due to the involvement of these signalling complexes in cancer, these new findings are important and may enable discovery of next-generation therapeutics.

The proof-or-principle experiment on SKP/SMP complex disruption using inhibitors presented in the manuscript is interesting, though new compounds would have to be developed to target un-mutated MRAS. It is, however, understandable that this exceeds the scope of the current work.

The paper is well written and reports interesting new findings that advance our understanding of the assembly of RAS-signalling complexes and pave the way for the discovery of future cancer therapeutics. As such, the paper is a possible candidate for publication in Nature Communications. However, the comments below will need to be addressed in order for the paper to adhere to the technical standards established in the field. Some minor comments and questions listed below should also be addressed.

Cryo-EM validation: The authors should add the following items to the manuscript to support the accuracy of cryo-EM structure determination:

- FSC curves between the model and the cryo-EM map (model vs. map FSC), evaluated at FSC = 0.5.

We thank the reviewer for the suggestion. The FSC evaluated at 0.5 is now shown in Supplementary Figure 1.

- Analysis of the anisotropy of the cryo-EM map, e.g. using the 3D FSC validation server.

We thank the reviewer for the suggestion. For the analysis of the anisotropy in the cryo-EM map, we utilized the 3D FSC validation server to generate a histogram and directional FSC plot. We also utilized orientation diagnostics in cryoSPARC to examine the relative signal, visualized in a 2D azimuth-elevation chart and a 3D scatter plot, as well as the relative signal within the twelve regions of the viewing sphere. Both plots are shown in Supplementary Figure 1.

- Figures showing the cryo-EM map and the refined model for important sections of the structure or where atomic interactions have been deduced (e.g. Fig. 2c-e; Fig. 3d-i) to support the accuracy of the modelling.

We thank the reviewer for their comment. We have revised Figure 2c-e to display the map around the mutation instead of the van der Waals. For Figure 3, we now show the map around the key interfaces and residues as a separate new Supplementary Figure (Supplementary Figure 4).

X-ray structures: The ligand densities are convincing, indicating that high-quality electron density maps have been obtained (as expected at the stated resolutions). However, Table 2 shows that the Rfree values are virtually identical across the X-ray structures, even though one might have expected the higher-resolution structures to exhibit lower Rfree values because the structures can be built more accurately.

We carefully re-examined all four structures in response to the reviewer's suggestion. Refinement strategies for KRASwt(GDP)+MRTX1133 and MRASmut(GDP)+MRTX1133 were checked by PDB-REDO. For KRASwt(GDP)+MRTX1133 (1.40Å), we were using isotropic B-factor refinement with TLS, however, Hamilton tests (a way to test the significance in the improvement of R/Rfree from the inclusion of more parameters, in this case anisotropic vs isotropic B-factor refinement, PMID: 22505269) conclusively selected anisotropic B-factor refinement. Re-refinement of the KRASwt(GDP)+MRTX1133 structure with anisotropic refinement for protein and ligands and isotropic refinement for water has lowered R/Rfree to 16.0/19.8. The Ramachandran outliers have been fixed, and this structure has been replaced in the PDB, and Supplementary Table 3 has been updated. For MRASmut(GDP)+MRTX1133 (1.50Å), Hamilton tests conclusively selected isotropic B-factor refinement, but no improvement of R/Rfree was observed. The maps show no significant difference peaks, and all residues are modeled with no additional waters required. However, the higher R/Rfree may be due to poor electron density surrounding Switch-II of the second copy of MRASmut(GDP) in the asymmetric unit (chain B). We rebuilt regions of this structure (see our response to the next comment) and also replaced it in the PDB.

We agree with the reviewer's suggestion. If the data quality were nearly identical across the three datasets, the R and Rfree values would be expected to vary according to resolution. However, this is not the case here. Despite using similar refinement protocols, the similar Rfree values across structures of differing resolution likely reflect a combination of factors, including differences in data quality (e.g., completeness, redundancy, and $I/\sigma(I)$) and other parameters, which can reduce variation in Rfree despite differences in nominal resolution.

At the same time, the validation reports indicate somewhat high Ramachandran outlier statistics for some structures. Could the authors double check their full models to ensure that they are built as well as possible at the reported resolutions?

We thank the reviewer for this suggestion. We have rebuilt regions of the MRASmut(GDP)+MRTX1133 structure due to the few Ramachandran outliers. This has improved the Molprobity scores in the PDB report. The re-refined structure has been deposited

to the PDB, and Supplementary Table 3 has been updated. The outliers for the KRASwt(GDP)+MRTX1133 were fixed during the anisotropic B-factor refinement (described above) and have also been redeposited. We have also added Ramachandran details to Supplementary Table 3 for all structures.

Page 4, bottom: Are the cellular concentrations of these proteins high enough to allow complex formation, given the affinities cited, or is it possible that H/K/NRAS only gain access to ternary complex formation in the presence of affinity-increasing mutations?

Physiological total RAS levels are low-micromolar to sub-micromolar, and PP1C/SHOC2 are present at moderate abundance (PMID: 11923843; BioNumbers ID 100889). Given the substantially higher affinity of MRAS for SHOC2 and PP1C, as well as the concentrating effect of membrane localization, MRAS can nucleate ternary assembly at endogenous expression levels. It is possible that due to their lower affinity, wild-type H/K/NRAS are unable to form a stable ternary complex unless their local concentration, GTP-bound concentration, or affinity for SHOC2 and PP1C is increased (PMID: 35831509; PMID: 36175670). Oncogenic mutations in KRAS/HRAS/NRAS can substantially increase the fraction of GTP-bound RAS and, together with membrane localization, crowding, and clustering, can overcome the relatively weak affinities observed in vitro, making ternary assembly at the membrane more favorable than predicted from solution K_{DS} alone (PMID: 15860728; PMID: 31674905).

Page 7, top: Is the number of salt bridges and hydrogen bonds taken directly from PISA output or has this been re-evaluated visually using the molecular model? If the latter, what criteria were applied?

The number of salt bridges and hydrogen bonds was taken directly from PDBSum (<https://www.ebi.ac.uk/thornton-srv/software/PDBsum1/>). We have modified the sentence to "...SKP complex buries approximately 1000 Å² less surface area and forms 6 fewer salt bridges and 12 fewer hydrogen bonds compared to the SMP complex (as calculated by the structural analysis programs PISA and PDBSum),..." for clarification.

Page 18, near the bottom: Three sentences are present twice (referencing MOLREP, REFMAC refinement, and PISA).

We thank the reviewer for pointing out these duplicated sentences. They have now been removed.

Fig. 2a: Local resolution maps are typically coloured such that high resolution regions are blue and low-resolution regions are red (mirroring low/high b-factors from crystallography). The authors should also consider changing the colour palette used because the current spectrum (the default in Chimera) can cause difficulty for those with impaired colour vision (e.g. Cramer et al., Nat. Commun. 2020; <https://www.nature.com/articles/s41467-020-19160-7>).

We thank the reviewer for the suggestion. The local resolution map in Figure 2a has been updated with a color palette that is friendly for individuals with impaired color vision.

Fig. 4: Switch I should be labelled in the figure panels (like Switch II).

We have labelled Switch I in Figure 4 as the reviewer suggested.

Point-by-point response to reviewer's comments:

We sincerely appreciate the reviewers' thorough and constructive feedback. In this revised manuscript, we have addressed all concerns raised, resulting in an improved and more robust version. Below, we provide point-by-point responses to the reviewers' comments (highlighted in blue).

Reviewer #1 (Remarks to the Author):

The authors have addressed most concerns, as well as those raised by other reviewers. Substantial revisions have been made to the manuscript, including modifications to figures and text, with new data in response to reviewer requests. Major revisions include:

1. Inclusion of Compound 5 in the study, demonstrates its effect on SHOC2–RAS–PP1C complex activity toward a CRAF substrate via dephosphorylation assays.
2. Addition of the standard error of the mean values for all ITC data in the Table, along with the K_D values reported by others.
3. Addition of "titrant → cell" labels in all ITC figures.
4. Repetition of ITC experiments to obtain additional data points in the sigmoidal region.
5. Improved visualization through updated surface and cartoon models.
6. Execution of in vitro dephosphorylation assays (with or without MRTX1133) using SHOC2 and PP1CA with MRAS, MRASmut, or KRAS against a phosphopeptide of CRAF-CR2pS, with a new figure included.
7. Addition of a new figure showing Western blot analysis in RAS-less cells transfected with GTP-locked MRAS(Q71L) or MRASmut(Q71L).
8. Inclusion of FSC evaluation at 0.5 in the relevant figure.
9. Rebuilding of MRASmut(GDP)+MRTX1133 structural regions to address Ramachandran outliers, with updated Ramachandran statistics included in the Table for all rebuilt structures.

In addition to these major updates, the authors have implemented several minor textual and figure improvements throughout the manuscript. However, one point mentioned in the rebuttal that remains unaddressed in the main text is that the authors have not clarified whether the ITC replicates are biological or technical. This information should be added to the figure legends.

We apologize to both Reviewer #1 and Reviewer #5 for the oversight. We have now revised the Methods section to clearly state: "Technical duplicate isothermal titration calorimetry measurements were performed on a MicroCal PEAQ-ITC instrument." Additionally, "A representative run of each experiment is shown in the figures with the average K_D reported for technical duplicate experiments. All ITC parameters (K_D , ΔH and $-T\Delta S$) have been tabulated as averages with their ranges." All figure legends have been updated to indicate: "The K_D is calculated from two technical replicates."

Moreover, the justification for the large discrepancy in K_D values (previously 120 nM vs. 900 nM by ITC) remains unclear. The explanation provided does not fully resolve this issue.

We measured the K_D of SMP formation as 120 nM by SPR, using 150 mM NaCl and up to 1 μ M PP1C in the titration, a concentration at which PP1C remains stable in solution. For ITC, much higher PP1C concentrations (~35–50 μ M) are required; at these concentrations, PP1C aggregates and precipitates in 150 mM NaCl. To maintain solubility, we used 500 mM NaCl and 5% glycerol, conditions commonly used for PP1C in ITC (PMID: 37002212, 22284538, 27572260). Thus, the difference in K_D values arises from the higher ionic strength used in ITC measurements. Similarly, the SKP complex shows weakening from 700 nM by SPR (150 mM NaCl) to 7 μ M by ITC (500 mM NaCl). We have added a sentence in the Methods to clarify: "Previous SPR measurements were made in 150 mM sodium

chloride.” Furthermore, we have included “SMP and SKP affinities measured by ITC, conducted in 500 mM NaCl and 5% glycerol to maintain PP1C solubility, are higher (900 nM for SMP, 7 μM for SKP) than those measured earlier by SPR at 150 mM NaCl (120 nM for SMP, 0.7 μM for SKP), reflecting the effect of ionic strength rather than intrinsic affinity differences (see Methods for details).” in the main text as well.

Additionally, the authors’ statement that “all other instances refer to wild-type SKP” appears inconsistent, since the SKP complex structures reportedly include one mutation each in SHOC2, KRAS, and PP1C—thus, these do not represent wild-type SKP.

We have revised the manuscript so that the SHOC2(M173I)-KRAS(Q61R)-PP1CA(P50R) complex is now referred to as the “stabilized SKP,” while the wild-type SHOC2-KRAS-PP1C complex is referred to as SKP throughout the manuscript.

We have altered the final two sentences of the ‘Mechanistic insights into KRAS versus MRAS engagement with SHOC2 and PP1C’ section to “The comparative analysis suggests that the cumulative loss of buried surface area, salt bridges, hydrogen bonds, and van der Waals contacts in the stabilized SKP complex is not due to the stabilization mutations but sequence differences between MRAS and KRAS. This underlies the reduced stability and lower affinity of the wild-type SKP complex relative to the SMP complex.” to summarize these observations.

Aside from these minor concerns, the revised manuscript is significantly improved and otherwise ready for publication.

Reviewer #2 (Remarks to the Author):

Reviewer #3 (Remarks to the Author):

Reviewer #4 (Remarks to the Author):

I have carefully reviewed the revised manuscript and the rebuttal provided by the authors. I feel that the authors have made a good response to my original critiques with my main issue being that they perform in vitro de-phosphorylation assays, which the authors have done.

Reviewer #5 (Remarks to the Author):

Bonsor, Finci, et al. have revised their manuscript according to the comments of several anonymous reviewers. While the manuscript has been improved, additional changes will be needed before it can be considered for publication.

Main comments:

1) The authors have generally addressed the reviewers' comments in a suitable manner in their response to reviewers, but in some cases, the changes to the manuscript itself are rather minimal. This applies particularly to some valid issues raised by reviewer #1, e.g. points 3, 4.

For Point #3, we have revised the manuscript so that the SHOC2(M173I)-KRAS(Q61R)-PP1CA(P50R) complex is now referred to as the “stabilized SKP,” while the wild-type SHOC2-KRAS-PP1C complex is referred to as SKP throughout the manuscript. We have altered the final two sentences of the ‘Mechanistic insights into KRAS versus MRAS engagement with SHOC2 and PP1C’ section to “The comparative analysis suggests that the cumulative loss of buried surface area, salt bridges, hydrogen bonds, and van der Waals contacts in the stabilized SKP complex is not due to the stabilization mutations but sequence differences between MRAS and KRAS. This underlies the reduced stability and lower affinity of the wild-type SKP complex relative to the SMP complex.” to summarize these observations.

For Point #4, we measured the K_D of SMP formation as 120 nM by SPR, using 150 mM NaCl and up to 1 μ M PP1C in the titration, a concentration at which PP1C remains stable in solution. For ITC, much higher PP1C concentrations (~35–50 μ M) are required; at these concentrations, PP1C aggregates and precipitates in 150 mM NaCl. To maintain solubility, we used 500 mM NaCl and 5% glycerol, conditions commonly used for PP1C in ITC (PMID: 37002212, 22284538, 27572260). Thus, the difference in K_D values arises from the higher ionic strength used in ITC measurements. Similarly, the SKP complex shows weakening from 700 nM by SPR (150 mM NaCl) to 7 μ M by ITC (500 mM NaCl). We have added a sentence in the Methods to clarify: “Previous SPR measurements were made in 150 mM sodium chloride.” Furthermore, we have included “SMP and SKP affinities measured by ITC, conducted in 500 mM NaCl and 5% glycerol to maintain PP1C solubility, are higher (900 nM for SMP, 7 μ M for SKP) than those measured earlier by SPR at 150 mM NaCl (120 nM for SMP, 0.7 μ M for SKP), reflecting the effect of ionic strength rather than intrinsic affinity differences (see Methods for details).” in the main text as well.

2) Along the same lines: From the rebuttal: "All ITC measurements reported are technical replicates, and the standard error of the mean has been calculated for each measurement. This information has now been explicitly stated in the Materials and Methods." - I could not find the terms "technical replicate" or "standard error of the mean" anywhere in the manuscript text (the latter appears in a supplementary table). Could the authors specify where this is stated in the methods (maybe line numbers would help)?

We apologize to both Reviewer #1 and Reviewer #5 for the oversight. We have now revised the Methods section to clearly state: “Technical duplicate isothermal titration calorimetry measurements were performed on a MicroCal PEAQ-ITC instrument.” Additionally, “A representative run of each experiment is shown in the figures with the average K_D reported for technical duplicate experiments. All ITC parameters (K_D , ΔH and $-T\Delta S$) have been tabulated as averages with their ranges.” All figure legends have been updated to indicate: “The K_D is calculated from two technical replicates.”

3) The validation of the structure determined by cryo-EM requires further adjustments:

i) The local resolution estimation in Fig. 2a is uninformative - almost everything is blue (3 Å), nothing

is red (7 Å). The range of resolutions displayed within the color ramp should probably be 3 Å to 5 Å, not 3 Å to 7 Å.

We thank the reviewer for their comment. The revised local resolution color ramp in Fig. 2a now runs from 3 Å to 5 Å, resulting in a more informative figure.

ii) Supplementary Table 2: The model resolution determined by model vs. map FSC at FSC = 0.143 is meaningless. This value should be removed. The value at FSC = 0.5 should be retained, as this is the correct threshold.

We have removed the FSC=0.143 entry from Supplementary Table 2.

iii) The use of DeepEMhancer and other advanced post-processing programs is not a problem as such, but use of DeepEMhancer maps for structure refinement and validation (e.g. phenix.real_space_refine and model vs. map FSC) is not generally an accepted practice in the field. Can the authors confirm that a conventionally post-processed map was used for determination of the model vs. map FSC? If not, this should be changed and the updated curve/cutoff value reported.

We thank the reviewer for this suggestion. The volume was sharpened separately with both CryoSPARC B-factor-based sharpening and DeepEMhancer v0.13 using its wideTarget training model. We have now updated the model vs. map FSC with a B-Factor-sharpened map and have updated the table to reflect this.

iv) In the interest of transparency, the authors may want to consider disclosing the use of DeepEMhancer in the figure legends where the maps treated with this algorithm are shown, rather than just in Supplementary Table 2 and the methods section.

We thank the reviewer for this suggestion. In the interest of transparency, we have now indicated the use of DeepEMhancer maps in the figure legends.

Other minor comments:

The paper on 3D FSC validation (Tan et al., Nature Methods, 2017) should be cited.

The reference for the 3D FSC validation has now been cited.

Supplementary Fig. 10: Showing the compound in question (compound 5) would be helpful to the non-specialist reader.

We have added the skeletal structures of Compound5, MRTX1133, and RMC6236 to the relevant Supplementary figures.